# A Class of Algorithms for General Instrumental Variable Models

**Niki Kilbertus**[*]
Helmholtz AI

**Matt J. Kusner**
University College London
The Alan Turing Institute

**Ricardo Silva**
University College London
The Alan Turing Institute

## Abstract

Causal treatment effect estimation is a key problem that arises in a variety of real-world settings, from personalized medicine to governmental policy making. There has been a flurry of recent work in machine learning on estimating causal effects when one has access to an instrument. However, to achieve identifiability, they in general require one-size-fits-all assumptions such as an additive error model for the outcome. An alternative is partial identification, which provides bounds on the causal effect. Little exists in terms of bounding methods that can deal with the most general case, where the treatment itself can be continuous. Moreover, bounding methods generally do not allow for a continuum of assumptions on the shape of the causal effect that can smoothly trade off stronger background knowledge for more informative bounds. In this work, we provide a method for causal effect bounding in continuous distributions, leveraging recent advances in gradient-based methods for the optimization of computationally intractable objective functions. We demonstrate on a set of synthetic and real-world data that our bounds capture the causal effect when additive methods fail, providing a useful range of answers compatible with observation as opposed to relying on unwarranted structural assumptions.[1]

## 1 Introduction

Machine learning is becoming more and more prevalent in applications that inform actions to be taken in the physical world. To ensure robust and reliable performance, many settings require an understanding of the causal effects an action will have before it is taken. Often, the only available source of training data is observational, where the actions of interest were chosen by unknown criteria. One of the major obstacles to trustworthy causal effect estimation with observational data is the reliance on the strong, untestable assumption of *no unobserved confounding*. To avoid this, only in very specific settings (e.g., front-door adjustment, linear/additive instrumental variable regression) it is possible to allow for unobserved confounding and still identify the causal effect (Pearl, 2009). Outside of these settings, one can only hope to meaningfully bound the causal effect (Manski, 2007).

In many applications, we have one or few treatment variables $X$ and one outcome variable $Y$. Nearly all existing approaches to obtain meaningful bounds on the causal effect of $X$ on $Y$ impose constraints on how observed variables are related, in order to mitigate the influence of unobserved confounders. One of the most useful structural constraints is the existence of an observable *instrumental variable* (IV): a variable $Z$, not caused by $X$, whose relationship with $Y$ is entirely mediated by $X$, see Pearl (2009) for a graphical characterization. The existence of an IV can be used to derive upper (lower) bounds on causal effects of interest by maximizing (minimizing) those effects among all IV models compatible with the observable distribution. *In this work, we develop algorithms to compute these*

---

[*]Majority of work done while at Max Planck Institute for Intelligent Systems and University of Cambridge.

[1]Code available at `https://github.com/nikikilbertus/general-iv-models`.

*bounds on causal effects over "all" IV models compatible with the data in a general continuous setting.* Crucially, the space of "all" models *cannot* be arbitrary, but it can be made very flexible. Instead of forcing a user to adopt a model space with hard constraints, we will allow for choice from a continuum of model spaces. Our approach rewards background knowledge with tighter bounds and it is not tied to an a priori inflexible choice, such as additivity or monotonicity. It avoids the adoption of unwarranted structural assumptions under the premise that they are needed due to the lack of ways of expressing more refined domain knowledge. The burden of the trade-off is put explicitly on the practitioner, as opposed to embracing possibly crude approximations due to the limitations of identification strategies.

Eliciting constraints that characterize "the models compatible with data" under a causal directed acyclic graph (DAG) for discrete variables is an active field of study, with contributions from the machine learning, algebraic statistics, economics, and quantum mechanics literature. This has provided complete characterizations of equality (Evans, 2019; Tian & Pearl, 2002) and inequality (Wolfe et al., 2019; Navascues & Wolfe, 2019) constraints. Enumerating all inequality constraints is in general super-exponential in the number of observed variables, even for discrete causal models. However, this line of work typically solves a harder problem than is strictly required for bounding causal effects: they provide symbolic constraints obtained by eliminating all hidden variables. While the pioneering work of Balke & Pearl (1994) in the discrete setting also provides symbolic constraints via a super-exponential algorithm, it introduces constraints that match the observed marginals of a latent variable model against the observable distribution. Thereby it provides a connection to non-symbolic, stochastic approaches for evaluating integrals, which we develop in this work.

Our key observation is that we can leverage recent advances in efficient gradient and Monte Carlo-based optimization of computationally intractable objective functions to bound the causal effect directly. This can be done even in the setting where $X$ is continuous, where none of the literature described above applies. We do so by (a) parameterizing the space of causal responses to treatment $X$ such that we can incorporate further assumptions that lead to informative bounds; (b) using a Monte Carlo approximation to the integral over the distribution of possible responses to $X$, where the distribution itself must be parameterized carefully to incorporate the structural constraints of an IV DAG model. This allows us to optimize over the domain-dependent set of all plausible models that are consistent with observed data to find lower/upper bounds on the target causal effect.

In Section 2, we describe the general problem of using instrumental variables when treatment $X$ is continuous. Section 3 develops our representation of the causal model. In Section 4 we introduce a class of algorithms for solving the bounding problem and our suggested implementation. Section 5 provides several demonstrations of the advantages of our method.

## 2    Current Approaches and Their Limitations

Balke & Pearl (1994) focused on partial identification (bounding) of causal effects on binary discrete models. Angrist et al. (1996) studied identification of effects for a particular latent subclass of individuals also in the binary case. Meanwhile, the econometrics literature has focused on problems where the treatment $X$ is continuous (Newey & Powell, 2003; Blundell et al., 2007; Angrist & Pischke, 2008; Wooldridge, 2010; Darolles et al., 2011; Horowitz, 2011; Chen & Christensen, 2018; Lewis & Syrgkanis, 2018). This problem has recently received attention in machine learning, using techniques from deep learning (Hartford et al., 2017; Bennett et al., 2019) and kernel machines (Singh et al., 2019; Muandet et al., 2020). This literature assumes that the structural equation for $Y$ has a special form, such as having an additive error term $e_Y$, as in $Y = f(X) + e_Y$. The error term $e_Y$ is not caused by $X$, but need not be independent of it, introducing unobserved confounding. This assumption is also used in related contexts, such as in sensitivity analysis for counterfactual estimands, see Kilbertus et al. (2019) for a specific application in fairness.

Using the notation of Pearl (2009), the expected response under an intervention on $X$ at level $x$ is denoted by $\mathbb{E}[Y \mid do(x)]$, which in the model above boils down to $f(x)$. An *average treatment effect* (ATE) can be defined as a contrast of this expected response under two treatment levels, e.g., $f(x) - f(x')$. In the zero-mean additive error case, $\mathbb{E}[Y \mid z] = \int f(x)p(x \mid z)\,dx$. Under some regularity conditions, no function other than $f(\cdot)$ satisfies that integral equation. Since $\mathbb{E}[Y \mid z]$ and $p(x \mid z)$ can be both learned from data, this allows us to learn the ATE from observational data. This

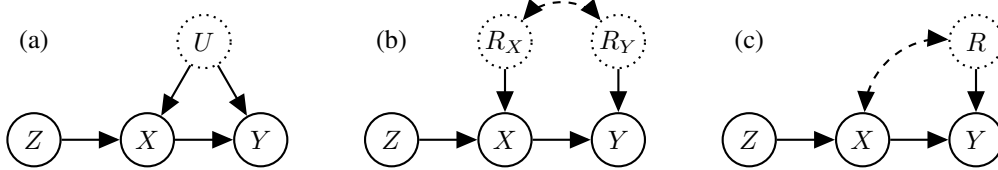

Figure 1: (a) An example of DAG compatible with $Z$ being an instrument for $X \to Y$, with hidden confounder $U$. (b) An equivalent representation using *response function* indices for deterministic functions $X = g_{R_X}(Z)$ and $Y = f_{R_Y}(X)$, with two random indexing variables $R_X$ and $R_Y$. (c) For the purposes of modeling $\mathbb{E}[Y \mid do(x)]$, it is enough to express the model in terms of $R := R_Y$ only.

is how the vast majority of recent work identifies the causal treatment effect in the IV model (Hartford et al., 2017; Bennett et al., 2019; Singh et al., 2019; Muandet et al., 2020).

The price paid for identification is that it seriously limits the applicability of these models. Diagnostic tests for the additivity assumption are not possible, as residuals $Y - f(X)$ can be arbitrarily associated with $X$ by assumption. On the other hand, without *any* restrictions on the structural equations, it is not only impossible to identify the causal effect of the IV model with a continuous treatment, but even bounds on the ATE are vacuous (Pearl, 1995; Bonet, 2001; Gunsilius, 2018, 2020). However, with relatively weak assumptions on the space of allowed structural equations, it is possible to achieve meaningful bounds on the causal effect (Gunsilius, 2020). It suffices that the equations for $X$ and $Y$ have a finite number of discontinuities. Gunsilius provides a theoretical framework for representation and estimation of bounds. Algorithmically, he proposes a truncated wavelet representation for the causal response and builds convex combinations of a sample of response functions to optimize IV bounds. Although it is an important proof of concept for the possibility of bounds for the general IV case with a strong theoretical motivation, we found that the method has frequent stability issues that are not easy to diagnose. We return to this in Appendix A.

Building on top of this work and some classical ideas first outlined by Balke & Pearl (1994), we propose an alternative formulation for finding bounds when both $X$ and $Y$ are continuous. Our technique flexibly parameterizes the causal response functions, while naturally encoding the structural IV constraints for compatibility with the observed data. We then leverage an augmented Lagrangian method that is tailored to non-convex optimization with inequality constraints. We demonstrate that our method matches estimation results of prior work in the additive setting, and gives meaningful bounds on the causal effect in general, non-additive models. Thereby, we follow a line of recent successes in various domains achieved by replacing previous intractable symbolic-combinatorial algorithms (Balke & Pearl, 1994; Wolfe et al., 2019; Drton et al., 2009) with a continuous program. One of our key contributions is to formulate bounds on true causal effects as well as their compatibility requirements as a smooth, constrained objective, for which we can leverage efficient gradient-based optimization techniques with Monte Carlo approximations.

## 3 Problem Setting

Following Pearl's Structural Causal Model (SCM) framework (Pearl, 2009), we assume the existence of structural equations and a (possibly infinite dimensional) unobserved exogenous process $U$,

$$X = g(Z, U) \quad \text{and} \quad Y = f(X, U). \tag{1}$$

We illustrate this situation in Figure 1(a). It assumes the usual requirements for the instrument $Z$ to be satisfied, namely (a) $Z \perp\!\!\!\perp U$, (b) $Z \not\!\perp\!\!\!\perp X$, and (c) $Z \perp\!\!\!\perp Y \mid \{X, U\}$.

### 3.1 Goal

The goal is to compute lower/upper bounds on $\mathbb{E}[Y \mid do(x^\star)]$ for any desired intervention level $x^\star$. Bounds on (conditional) ATEs can be derived, see also Appendix B. Intuitively, we put bounds on how $f(X, U)$ depends on $X$ by optimizing over "allowed" distributions of $U$. Which distributions are "allowed" is determined by observations, i.e., we only consider settings where marginalizing $U$ results in $p(x, y \mid z)$ for all $(x, y, z)$ in the support of the observational distribution. In fact, as pointed out by Palmer et al. (2011), it is enough to consider matching the marginals of the latent variable

model to the two conditional densities $p(x \mid z)$ and $p(y \mid z)^2$. Informally, *among all possible structural equations* $\{g, f\}$ *and distributions over* $U$ *that reproduce the estimated densities* $\{\hat{p}(x \mid z), \hat{p}(y \mid z)\}$, *we find estimates of the minimum and maximum expected outcomes under intervention.*

**Response functions.** The main idea of Balke & Pearl (1994) is to express structural equations in terms of *response functions*: labeling (and possibly clustering) states of $U$ according to the implied functional relationship between the observed variable and its direct causes. These $U$ states are mapped to a particular level of an index variable $R$. For instance, if $Y = f(X, U) = \lambda_1 X + \lambda_2 X U_1 + U_2$, a two-dimensional $U$ space in a linear, non-additive outcome function, we have that $f(x, u) = \lambda_1 x + \lambda_2 x$ for $u_1 = 1$, $u_2 = 0$. We can define an implicit arbitrary value $r$ such that $f_r(x) = \lambda_r x$, $\lambda_r = \lambda_1 + \lambda_2$, the value "$r$" being an alias for $(1, 0)$ in the space of the confounders. The advantage of this representation is that we can think of a distribution over $R$ as a distribution over functions of $X$ alone. Otherwise we would need to deal with interactions between $U$ and $X$ on top of a distribution over $U$, itself of unclear dimensionality. In contrast, the dimensionality of $R$ is the one implied by the particular function space adopted. Gunsilius (2020) provides a more thorough discussion of the role of response functions corresponding to a possibly infinite-dimensional $U$. Figure 1(b) shows a graphical representation of a system parameterized by response function indices $R_X$ and $R_Y$, with a bi-directed edge indicating possible association between the two. In what follows, as there will be no explicit need for $R_X$, the causal DAG corresponding to our counterfactual model is shown in Figure 1(c)[3]. This itself departs from Balke & Pearl (1994) and Gunsilius (2020), having the advantage of simplifying the optimization and not assuming counterfactuals for $X$ (which will not exist if $Z$ is not a cause of $X$ but just confounded with it). Furthermore, focusing on $\{p(x \mid z), p(y \mid z)\}$ instead of $p(x, y \mid z)$ does not require simultaneous measurements of $X$ and $Y$ (Palmer et al., 2011), see Appendix G for the latter. Within this framework, we can rewrite the optimization over allowed distributions of $U$ into an optimization over allowed distributions of response functions for $Y$.

Without restrictions on the function space, non-trivial inference is impossible (Pearl, 1995; Bonet, 2001; Gunsilius, 2018). In our proposed class of solutions, we will adopt a parametric response function space: each response type $r$ corresponds to some parameter value $\theta_r \in \Theta \subset \mathbb{R}^K$ for some finite $K$. We write $f_r(x) := f_{\theta_r}(x)$. Going forward, we will simply use $\theta$ to denote a specific response type and drop the index $r$. While our method works for any differentiable $f_\theta$, we will focus on linear combinations of a set of basis functions $\{\psi_k : \mathbb{R} \to \mathbb{R}\}_{k \in [K]}$[4] with coefficients $\theta \in \Theta$:

$$f_\theta(x) := \sum_{k=1}^{K} \theta_k \, \psi_k(x). \tag{2}$$

We propose to optimize over distributions $p_{\mathcal{M}}(\theta)$ of the response function parameters $\theta$ in the unknown causal model $\mathcal{M}$, subject to the observed marginal of the model, $\int p_{\mathcal{M}}(x, y \mid z, \theta) p_{\mathcal{M}}(\theta) \, d\theta$, matching the corresponding (estimated) marginals $p(y \mid z)$ and $p(x \mid z)$. Notice that $\theta \perp\!\!\!\perp Z$ is implied by $Z \perp\!\!\!\perp U$ in the original formulation in terms of exogenous variables $U$. We assume a parametric form for $p_{\mathcal{M}}(\theta)$ via parameters $\eta \in \mathbb{R}^d$, denoted by $p_\eta(\theta)$. We propose to use function families for $p_\eta(\theta)$ that allow for practically low-variance Monte-Carlo gradient estimation via the reparameterization trick (Kingma & Welling, 2014) to learn $\eta$ — more in Section 3.2.

**Objective.** An upper bound for the expected outcome under intervention can be directly written as

$$\max_\eta \mathbb{E}[Y \mid do(x^\star)] = \max_\eta \int f_\theta(x^\star) \, p_\eta(\theta) \, d\theta. \tag{3}$$

A lower bound can be found analogously by the minimization problem. When optimizing eq. (3) constrained by $p(y \mid z)$ and $p(x \mid z)$ in the sequel, it will be necessary to define $p_\eta(x, \theta \mid z)$.[5] In particular, $\int p_\eta(x, \theta \mid z) \, dx = p_\eta(\theta \mid z) = p_\eta(\theta)$. The last equality will be enforced in the encoding of $p_\eta(x, \theta \mid z)$, as we need $Z \perp\!\!\!\perp \theta$ even if $Z \not\perp\!\!\!\perp \theta \mid X$. This encoding is introduced in Section 3.2, which will also allow us to easily match the marginal $p(x \mid z)$. In Section 3.3, we construct constraints for the optimization so that the marginal of $Y$ given $Z$ in $\mathcal{M}$ matches the model-free $p(y \mid z)$.

## 3.2 Matching $p(x \mid z)$ and Enforcing $Z \perp\!\!\!\perp U$

Instead of formulating the criterion of preserving the observed marginal $p(x \mid z)$ as a constraint in the optimization problem, we bake it directly into our model.[6] To accomplish that, we factor $p_\eta(x, \theta \mid z)$ as $p(x \mid z) p_\eta(\theta \mid x, z)$. The first factor is identified from the observed data and we can thus force our model to match it. The second factor must be constructed so as to enforce marginal independence between $\theta$ and $Z$ (as required by $Z \perp\!\!\!\perp U$). We achieve that by parameterizing it by a copula density $c_\eta(\cdot)$ that takes univariate CDFs $F(\cdot)$, which uniquely define the distributions, as inputs,

$$p_\eta(\theta \mid x, z) := c_\eta(F(x \mid z), F_\eta(\theta_1), \ldots, F_\eta(\theta_K)) \prod_{k=1}^{K} p_\eta(\theta_k). \tag{4}$$

Here we assume that each component $\theta_k$ of $\theta$ has a Gaussian marginal density with mean $\mu_k$ and variance $\sigma_k^2$, i.e., $p_\eta(\theta_k) = \mathcal{N}(\theta_k; \mu_k, \sigma_k^2)$. Moreover, assuming $c_\eta$ is a multivariate Gaussian copula density requires a correlation matrix $S \in \mathbb{R}^{(K+1) \times (K+1)}$ for which we only keep a Cholesky factor $L$ without further constraints, rescaling $L^\mathsf{T} L$ to have a diagonal of 1s. Our full set of parameters is

$$\eta := \{\mu_1, \ln(\sigma_1^2), \ldots, \mu_K, \ln(\sigma_K^2), L\} \in \mathbb{R}^{K(K+1)/2 + 2K}.$$

## 3.3 Matching $p(y \mid z)$

In the continuous output case, our parameterization implies the following set of integral equations

$$\Pr(Y \le y \mid Z = z) = \int \mathbf{1}(f_\theta(x) \le y) \, p_\eta(x, \theta \mid z) \, dx \, d\theta, \tag{5}$$

for all $y \in \mathcal{Y}, z \in \mathcal{Z}$, the respective sample spaces of $Y$ and $Z$, where $\mathbf{1}(\cdot)$ is the indicator function. These constraints immediately introduce two difficulties. First, we have an infinite number of constraints to satisfy. Second, the right-hand side involves integrating non-continuous indicator functions, which poses a problem for smooth gradient-based optimization with respect to $\eta$.[7]

To circumvent these issues, we first choose a finite grid $\{z^{(m)}\}_{m=1}^{M} \subset \mathcal{Z}$ of size $M \in \mathbb{N}$, instead of conditioning on all values in $\mathcal{Z}$. We compute $z^{(m)}$ from a uniform grid on the CDF $F_Z$ of $Z$, i.e., $z^{(m)} := F_Z^{-1}(m/M+1)$ for $m \in [M]$. Second, to avoid the integration of non-continuous indicator functions, we can express the constraints of eq. (5) in terms of expectations over a dictionary of $L$ basis functions $\{\phi_l\}_{l=1}^{L}$. This leads to the following constraints for $p(y \mid z)$:

$$\mathbb{E}[\phi_l(Y) \mid z^{(m)}] = \int \phi_l(f_\theta(x)) \, p_\eta(x, \theta \mid z^{(m)}) \, dx \, d\theta \quad \text{for all } l \in [L], m \in [M]. \tag{6}$$

This idea borrows from mean embeddings, where one can reconstruct $p(y \mid z)$ from an infinite dictionary sampled at infinitely many points in $\mathcal{Z}$ (Singh et al., 2019). In this work, we choose an even simpler approach and only constrain moments like mean and variance $\phi_1(Y) := \mathbb{E}[Y]$, $\phi_2(Y) := \mathbb{V}[Y], \ldots$. Crucially, we note that *our approximations can only relax the constraints*, i.e., the optima may result in looser bounds compared to the full constraint set, *but not invalid bounds*, barring bad local optima as well as Monte Carlo and estimation errors.

# 4 Optimization Strategy

Here we state our final non-convex, yet smooth, constrained optimization problem:

$$
\begin{aligned}
\text{objective:} \qquad & o_{x^\star}(\eta) := \int f_\theta(x^\star) \, p_\eta(\theta) \, d\theta \\
\text{constraint LHS:} \qquad & \text{LHS}_{m,l} := \mathbb{E}[\phi_l(Y) \mid z^{(m)}] \\
\text{constraint RHS:} \qquad & \text{RHS}_{m,l}(\eta) := \int \phi_l(f_\theta(x)) \, p_\eta(x, \theta \mid z^{(m)}) \, dx \, d\theta \\
\textbf{opt. problem:} \qquad & \min_\eta / \max_\eta o_{x^\star}(\eta) \quad \text{s.t.} \quad \text{LHS}_{m,l} = \text{RHS}_{m,l}(\eta) \text{ for all } m \in [M], l \in [L]
\end{aligned}
$$

Here, $\min$ and $\max$ give the lower and upper bound respectively. In this section we describe how to tackle the optimization with an augmented Lagrangian strategy (Nocedal & Wright, 2006) and how to estimate all quantities from observed data. Algorithm D in Appendix D describes the full procedure.

## 4.1  Augmented Lagrangian Strategy

We can think of the left-hand side LHS as target values, estimated once up front from observed data. The right-hand side RHS is estimated repeatedly using (samples from) our model $p_\eta(x, \theta \,|\, z^{(m)})$ during optimization. For notational simplicity, we will often "flatten" the indices $m$ and $l$ into a single index $l \in [M \cdot L]$. Since LHS is subject to misspecification and estimation error, we introduce positive tolerance variables $b \in \mathbb{R}_{>0}^{M \cdot L}$, relaxing equality constraints into inequality constraints

$$c_l(\eta) := b_l - |\mathrm{LHS}_l - \mathrm{RHS}_l(\eta)| \geq 0, \quad \text{with } b_l := \max\{\epsilon_{\mathrm{abs}}, \epsilon_{\mathrm{rel}} \cdot |\mathrm{LHS}_l|\},$$

for fixed absolute and relative tolerances $\epsilon_{\mathrm{abs}}, \epsilon_{\mathrm{rel}} > 0$. The constraint $c_l(\eta)$ is satisfied if $\mathrm{RHS}_l(\eta)$ is *either* within a fraction $\epsilon_{\mathrm{rel}}$ of $\mathrm{LHS}_l$ *or* within $\epsilon_{\mathrm{abs}}$ of $\mathrm{LHS}_l$ in absolute difference. The absolute tolerance is useful when LHS is close to zero. The exact constraints are recovered as $\epsilon_{\mathrm{abs}}, \epsilon_{\mathrm{rel}} \to 0$. Again, the introduced tolerance can only make the obtained bounds looser, not invalid.

We consider an inequality-constrained version of the augmented Lagrangian approach with Lagrange multipliers $\lambda \in \mathbb{R}^{M \cdot L}$ (detailed in Section 17.4 of Nocedal & Wright (2006)). Specifically, the Lagrangian we aim to minimize with respect to $\eta$ is:

$$\mathcal{L}(\eta, \lambda, \tau) := \pm o_{x^\star}(\eta) + \sum_{l=1}^{M \cdot L} \begin{cases} -\lambda_l c_l(\eta) + \frac{\tau c_l(\eta)^2}{2} & \text{if } \tau c_l(\eta) \leq \lambda_l, \\ -\frac{\lambda_l^2}{2\tau} & \text{otherwise,} \end{cases} \tag{7}$$

where $+/-$ is used for the lower/upper bound and $\tau$ is a temperature parameter, which is increased throughout the optimization procedure. Given an approximate minimum $\eta$ of this subproblem, we then update $\lambda$ and $\tau$ according to $\lambda_l \leftarrow \max\{0, \lambda_l - \tau c_l(\eta)\}$ and $\tau \leftarrow \alpha \cdot \tau$ for all $l \in [M \cdot L]$ and a fixed $\alpha > 1$. The overall strategy is to iterate between minimizing eq. (7) and updating $\lambda_l$ and $\tau$.

## 4.2  Empirical Estimation and Implementation Choices

For a dataset $\mathcal{D} = \{(z_i, x_i, y_i)\}_{i=1}^N \subset \mathbb{R}^3$, we describe our method in Algorithm D in Appendix D.

**Pre-processing.** As a first step, we whiten the data (subtract mean, divide by variance). Then, we interpolate the CDF $\hat{F}_Z$ of $\{z_i\}_{i=1}^N$ to compute the grid points $z^{(m)}$. Next, we assign each observation to a grid point via $\mathrm{bin}(i) := \max\{\arg\min_{m \in [M]} |z_i - z^{(m)}|\}$ for $i \in [N]$, i.e., each datapoint is assigned to the gridpoint that is closest to its $z$-value (higher bin for ties). Given $M, L$ and $\phi_l$, we can estimate $\mathrm{LHS}_{m,l} := \mathbb{E}[\phi_l(Y) \,|\, z^{(m)}] \approx \frac{1}{|\mathrm{bin}^{-1}(m)|} \sum_{i \in \mathrm{bin}^{-1}(m)} \phi_l(y_i)$, which remain unchanged throughout the optimization. This allows us to fix the tolerances $b = \max\{\epsilon_{\mathrm{abs}}, \epsilon_{\mathrm{rel}} \mathrm{LHS}\}$. Finally, we obtain a single batch of examples from $X \,|\, z^{(m)}$ of size $B \in \mathbb{N}$, which we will also reuse throughout the optimization via inverse CDF sampling $\hat{x}_j^{(m)} = \hat{F}_{X \,|\, z^{(m)}}^{-1}\big((j-1)/(B-1)\big)$ for $j \in [B], m \in [M]$. Here, $\hat{F}_{X \,|\, z^{(m)}}$ is the CDF of $\{x_i\}_{i \in \mathrm{bin}^{-1}(m)}$.

**Monte Carlo estimation[8].** To minimize the Lagrangian, we use stochastic gradient descent (SGD). Therefore, we need to compute (estimates for) $\nabla_\eta o_{x^\star}(\eta), \nabla_\eta c_l(\eta)$, where the latter boils down to $\nabla_\eta \mathrm{RHS}_{m,l}(\eta)$. In practice, we compute Monte Carlo estimates of $o_{x^\star}(\eta)$ and $\mathrm{RHS}_{m,l}(\eta)$ and use automatic differentiation, e.g., using JAX (Bradbury et al., 2018), to get the gradients. If we had a batch of independent samples $\theta^{(j)} \sim p_\eta(\theta)$ of size $B$, we could estimate the objective eq. (3) for a given $\eta$ via $\mathbb{E}[Y \,|\, do(x^\star)] \approx \frac{1}{B} \sum_{j=1}^B f_{\theta^{(j)}}(x^\star)$. Similarly, with i.i.d. samples $\theta^{(j)} \sim p_\eta(\theta \,|\, z^{(m)})$ we can estimate $\mathrm{RHS}_{m,l}$ in eq. (6) as $\mathrm{RHS}_{m,l}(\eta) \approx \frac{1}{B} \sum_{j=1}^B \phi_l\big(f_{\theta^{(j)}}(\hat{x}_j^{(m)})\big)$. Hence, the last missing piece is to sample from eq. (4) in a fashion that maintains differentiability w.r.t. $\eta$. We follow the standard procedure to sample from a Gaussian copula for the parameters $\theta^{(j)}$, with the additional restriction to preserve the pre-computed sample $\hat{x}$. Algorithm 2 in Appendix D describes the sampling process from $p_\eta(\theta, X \,|\, z^{(m)})$ as defined in Section 3.2 in detail. The output is a $(K+1) \times B$-matrix, where the first row contains $B$ independent $X$-samples and the remaining $K$ rows are the components of $\theta \in \mathbb{R}^K$. We pool samples from all $z^{(m)}$ to obtain samples from $p_\eta(\theta)$. By change of variables, the parameters $\eta = (\mu, \sigma^2, L)$ enter in a differentiable fashion (c.f. reparameterization trick (Kingma & Welling, 2014)). We initialize $\eta$ randomly, described in detail in Appendix D.4.

**Response functions.** For the family of response functions we first consider polynomials, i.e., $\psi_k(x) = x^{k-1}$ for $k \in [K]$. We will specifically focus on linear ($K = 2$), quadratic ($K = 3$), and cubic ($K = 4$) response functions. Second, we consider *neural basis functions (MLP)*, where we fit a multi-layer perceptron with $K$ neurons in the last hidden layer to the observed data $\{(x_i, y_i)\}_{i \in N}$ and take $\psi_k(x)$ to be the activation of the $k$-th neuron in the last hidden layer. We describe the details as well as an additional choice based on Gaussian process basis functions in Appendix E.

The choice of polynomials mainly illustrates a type of sensitivity analysis: we will contrast how bounds change when moving from a linear to quadratic, then quadratic to cubic and learned MLP representations. Recall that finite linear combinations of basis functions can arbitrarily approximate infinite-dimensional function spaces. The practitioner chooses its plausible complexity and pays the corresponding price in terms of looseness of bounds. For instance, we can add as many knot positions for a regression splines procedure as we want to get arbitrarily close to nonparametric function spaces. There is no concern for overfitting, given that data plays a role only via the estimation of $p(x, y \mid z)$ or of particular black-box expectations (Appendix G). We emphasize that *having a class of algorithms that allows for controlling the complexity of the function space is an asset, not a liability.* Knowledge of functional constraints is useful even in a non-causal setting (Gupta et al., 2020). The linear basis formulation can be as flexible as needed, while allowing for shape and smoothness assumptions that are more expressive than all-or-nothing assumptions about, say, missing edges or additivity. In Appendix F, we discuss an alternative based on discretization of $X$ combined with the off-the-shelf use of Balke & Pearl (1994). We demonstrate how in several ways that is just a *less* flexible family of response functions than the approach discussed here, although see Appendix B for a discussion on making $p_\eta(\theta \mid x, z)$ also more flexible than the implementation discussed here.

## 5  Experimental Results

We evaluate our method on a variety of synthetic and real datasets. In all experiments, we report the results of two stage least squares (**2SLS** - - -) and kernel instrumental variable regression (**KIV** -·-·) (Singh et al., 2019). Note that both methods assume additive noise and provide point estimates for expected outcomes under a given treatment. The KIV implementation by Singh et al. (2019) comes as an off-the-shelf method with internal heuristics for tuning hyperparameters. For our method, we show **lower** (·· ✕ ··) and **upper** (·· ✕ ··) bounds computed individually for multiple values of $x^\star \in \mathbb{R}$. The transparency of these lines indicates the tolerances $\epsilon_{\mathrm{abs}}, \epsilon_{\mathrm{rel}}$, where more transparency corresponds to larger tolerances. Missing bounds at an $x^\star$ indicate that the constraints could not be satisfied in the optimization. In the synthetic settings, we also show the **true causal effect** $\mathbb{E}[Y \mid do(X = x^\star)]$ (——).

Finally, we highlight that there are multiple possible causal effects compatible with the data (which our method aims to bound). To do so, we fit a latent variable model of the form shown in Figure 1(a) to the data, with $U \mid Z, X, Y \sim \mathcal{N}(\mu(Z, X, Y), \sigma^2(Z, X, Y))$ where $\mu, \sigma^2$ as well as $\mathbb{E}[X \mid Z, U]$ are parameterized by neural networks. We ensure that the form of $\mathbb{E}[Y \mid X, U]$ matches our assumptions on the function form of the response family (i.e., either polynomials of fixed degree in $X$, or neural networks). We then optimize the evidence lower bound following standard techniques (Kingma & Welling, 2014), see Appendix H. We fit multiple models with different random initializations and compute **the implied causal effect** of $X$ on $Y$ for each one, shown as multiple thin gray lines (——). We report results for additional datasets as well as how our method performs in the small data regime in Appendix I. All experiments use a single set of hyperparameters, see Appendix I.1.

**Linear Gaussian case.** First, we test our method in a synthetic linear Gaussian scenario, where instrument, confounder, and noises $Z, C, e_X, e_Y$ are independent standard Gaussian variables. We consider two settings of the form $X = g(Z, C, e_X) := \alpha Z + \beta C + e_X$ and $Y = f(X, C, e_Y) := X - 6C + e_Y$, with $\alpha, \beta \in \{(0.5, 3), (3, 0.5)\}$. The two settings of coefficients $\alpha, \beta$ describe a weak instrument with strong confounding and a strong instrument with weak confounding respectively. The first two rows of Figure 2 show our bounds in these settings for linear, quadratic and MLP response functions. Because these scenarios satisfy all theoretical assumptions of 2SLS and KIV, 2SLS (- - -) reliably recovers the true causal effect, which is simply $\mathbb{E}[Y \mid do(X = x^\star)] = x^\star$. For a weak instrument, KIV (-·-·) fails by reverting to its prior mean 0 everywhere, whereas it matches the true effect in data rich regions in the second setting with weak confounding.[9]

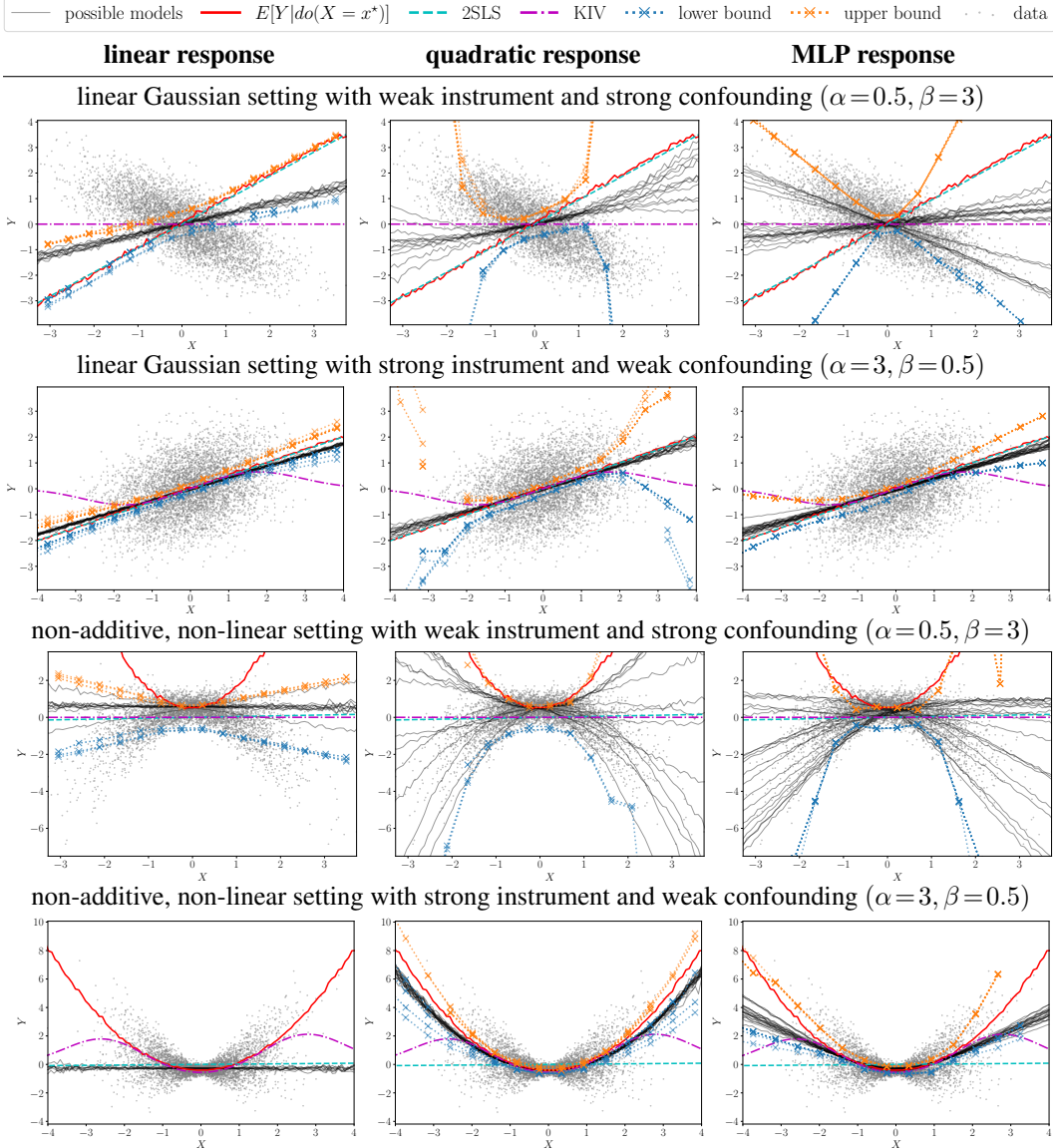

Figure 2: Results for synthetic datasets (linear Gaussian and non-linear, non-additive) for a weak and strong instrument respectively. Columns correspond to different response function families.

We observe that the true causal effect (——) is always within our bounds ($\cdot\cdot\times\cdot\cdot$, $\cdot\cdot\times\cdot\cdot$). Moreover, our bounds also contain most of the "other possible models" that could explain the data (——), showing that they are highly informative, without being more confident than warranted. As expected, our bounds get looser as we increase the flexibility of the response functions (linear, quadratic, MLP from columns 1-3). In particular, allowing for flexible MLP responses (column 3), our bounds are rightfully loose for strong confounding. As confounding weakens and the instrument strengthens (in the second row) the gap between our bounds gets narrower.

**Non-additive, non-linear case.** Our next synthetic setting is non-linear and violates the additivity assumption. Again, the treatment is given by $X = \alpha Z + \beta C + e_X$ with the same set of coefficients $\alpha, \beta$ as for the linear setting. The outcome is non-linear and non-additive $Y = 0.3 X^2 - 1.5 X C + e_Y$ with a true effect of $\mathbb{E}[Y \mid do(X = x^\star)] = 0.3 (x^\star)^2$. The bottom two rows of Figure 2 show our results for this setting. Since additivity is violated (due to the $X C$-term) and the effect is non-linear, 2SLS fails. Without additivity, KIV also fails for strong confounding, but captures the true effect well in data rich regions when the instrument is strong and confounding is weak. The strongly confounded case (row 3) highlights the effect of the choice of response functions. Wrongly assuming linear

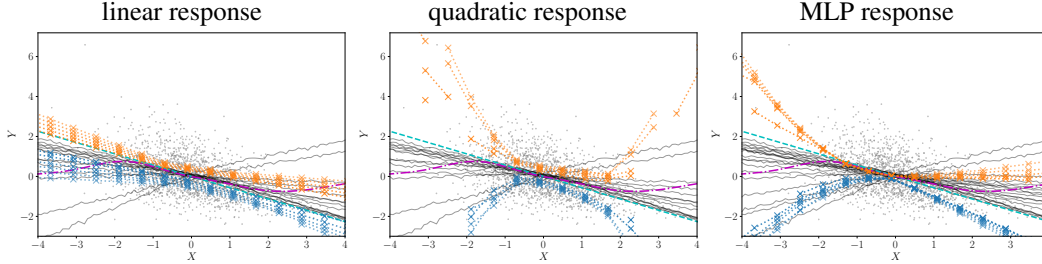

Figure 3: Results on the expenditure dataset for different response function families.

response functions, our bounds rule out the true effect (row 3, column 3). However, they capture the implied causal effects from possible compatible linear models. As we allow for more flexible response functions capable of describing the true effect, our bounds are extremely conservative (row 3, columns 2 & 3) as they should be, indicated by the effects from other compatible models. In the strong instrument, weak confounding case (row 4), our bounds become narrower to the point of essentially identifying the true effect for adequate response functions (column 2). Here, linear response functions cannot explain the data anymore, indicated by missing bounds (row 4, column 1).

**Expenditure data.** We now turn to a real dataset from a 1995/96 survey on family expenditure in the UK (Office for National Statistics, 2000). This dataset has been used by Gunsilius (2020) and previously (Blundell et al., 2007; Imbens & Newey, 2009) for 1994/95 data. The outcome of interest is the share of expenditure on food. The treatment is the log of the total expenditure and the instrument is gross earnings of the head of the household. All three variables are continuous, relations cannot be expected to be linear, and we cannot exclude unobserved confounding (Gunsilius, 2020), making this a good test case for our method. We describe the data in more detail in Appendix I.3. Figure 3 shows that our bounds provide useful information about both the sign and magnitude of the causal effect and gracefully capture the increasing uncertainty as we allow for more flexible response functions. Moreover, they include most of the possible effects from latent variable models indicating that they are not overly restrictive. The few curves that escaped our bounds correspond to situations where the latent variable model fit was suboptimal in terms of local likelihood and hence may be an artifact of the latent variable model training procedure.

## 6   Conclusion

We have proposed a class of algorithms for computing bounds on causal effects by exploiting modern optimization machinery. While this addresses an important source of uncertainty in causal inference — partial identifiability as opposed to full identifiability — there is also statistical uncertainty: confidence or credible intervals for the *bounds* themselves (Imbens & Manski, 2004). Clearly this is an important matter to be addressed in future work, and the black-box approach of Silva & Evans (2016) provides some directions for credible intervals. There are also considerations about the parameterization of $p_\eta(\theta \mid x, z)$ and how possible pre-treatment covariates can be non-trivially used in the model. We defer these considerations to Appendix B. Other parameterizations of the IV model, such as the one by Zhang & Bareinboim (2020) can lead to alternative algorithms and ways of expressing assumptions.

One could also use the same ideas to test whether an IV model is valid, another common use of latent variable causal models (e.g., Wolfe et al., 2019). We assumed that the model was correct. Model falsification can still be done, which will happen when the optimization fails to find a solution (Silva & Evans, 2016), and observed in some of the experiments reported. Specializing methods for testing models instead of deriving bounds is an interesting direction for future work.

Finally, we foresee our ideas as ways of liberating causal modeling to accommodate "softer," more general constraints than conditional independence statements. For instance, as described by Silva & Evans (2016), there is no need to assume any sparsity in a causal DAG, as long as we know that some edges are "weak" (in a technical sense) so that, e.g., edge $Z \to Y$ is allowed, but its influence on $Y$ is not arbitrary. How to do that in a computationally feasible way remains a challenge, but the possibility of complementing causal inference based on sparse DAGs, such as the do-calculus of Pearl (2009), with the sledgehammer of modern continuous optimization, is an attractive prospect.

## Broader Impact

Cause effect estimation is crucial in many areas where data-driven decisions may be desirable such as healthcare, governance or economics. These settings commonly share the characteristic that experimentation with randomized actions is unethical, infeasible or simply impossible. One of the promises of causal inference is to provide useful insights into the consequences of hypothetical actions based on observational data. However, causal inference is inherently based on assumptions, which are often untestable. Even a slight violation of the assumptions may lead to drastically different conclusions, potentially changing the desired course of action. Especially in high-stakes scenarios, it is thus indispensable to thoroughly challenge these assumptions.

This work offers a technique to formalize such a challenge of standard assumptions in continuous IV models. It can thus help inform highly-influential decisions. One important characteristic of our method is that while it can provide informative bounds under certain assumptions on the functional form of effects, the bounds will widen as less prior information supporting such assumptions is available. We can view this as a way of deferring judgment until stricter assumptions have been assessed and verified.

Since our algorithms are causal inference methods, they requires assumptions too. Therefore, our method also requires a careful assessment of these assumptions by domain-experts and practitioners. In addition, as we are optimizing a non-convex problem with local methods, we have no theoretical guarantee of correctness of our bounds. Hence, if wrong assumptions for our model are accepted prematurely, or our optimization strategy fails to find global optima, our method may wrongly inform decisions. If these are high-stakes decisions, then wrong decisions can have significant negative consequences (e.g., a decision not to treat a patient that should be treated). If the data that this model is trained on is biased against certain groups (e.g., different sexes, races, genders) this model will replicate those biases. We believe a fruitful approach towards making our model more sensitive to uncertainties due to structurally-biased, unrepresentative data, is to learn how to derive, then inflate (to account for bias) uncertainty estimates for our bounds.

## Acknowledgments and Disclosure of Funding

We thank Florian Gunsilius for useful discussions, providing code for his method and explaining how to prepare the Family Expenditure Survey dataset. We are grateful to Robin Evans, Arthur Gretton, Jiri Hron, Paul Rubenstein, and Rahul Singh for useful discussions and feedback. MK and RS acknowledge support from the The Alan Turing Institute under EPSRC grant EP/N510129/1. This work was partially done while RS was on a sabbatical at the Department of Statistics, University of Oxford.

## Footnotes

[2]In Appendix G, we discuss the case where we match $p(y \mid x, z)$, which can further tighten bounds with some computational advantages and disadvantages compared to $p(y \mid z)$.

[3]It is also possible to represent only $R_X$ and drop $R_Y$. Zhang & Bareinboim (2020) do this in a way that provides a new view of the discrete treatment case.

[4]We use the notation $[K] := \{1, \ldots, K\}$ for $K \in \mathbb{N}_{>0}$.

[5]We abuse notation slightly by expanding the definition of $\eta$ to simultaneously signify all parameters specifying this joint distribution, as well as individual parameters specific to certain factors of the joint.

[6] A full discussion on the construction and implications of such assumptions is given in Appendix B.

[7] We discuss discrete outcomes or discrete features, which could also lead to discontinuous $f_\theta$ in Appendix C.

[8]To be clear, *for the particular choice of $\phi$ and $p_\eta(\cdot \,|\, \cdot)$ in our experiments, Monte Carlo is not necessary*, see Appendix G. All experiments are done using Monte Carlo to test its suitability for general use.

[9]We provide more details on this failure mode of KIV in Appendix J.

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
