[Supplementary Material]

# A  Gunsilius's Algorithm

Gunsilius (2020) provides a theoretical framework for minimal conditions for a continuous IV model to imply non-trivial bounds (that is, bounds tighter that what can be obtained by just assuming that the density function $p(x, y \mid z)$ exists). That work also introduces two variations of an algorithm for fitting bounds.

The basic version consists of first sampling $l$ response functions $f_{R_x}(\cdot)$ and $f_{R_y}(\cdot)$ from a distribution over functions – in the experiments described, a Gaussian process evaluated on a grid in the respective spaces. The final distribution is reweighted combination of the pre-sampled $l$ response functions with weights $\mu$ playing the role of the decision variables to be optimized. Hence, by construction, the space of distributions in the response function space is absolutely continuous with respect to the pre-defined Gaussian process. The constraints are defined by approximating an estimate of the bivariate CDF $F(x, y \mid z)$ on a grid of values, which are approximately constrained to match the model implied CDF in a $L_2$ sense. Large deviance bounds are then used to show the (intuitive) result that this approximation is a probably approximately correct formulation of the original optimization problem.

One issue with this algorithm is that $l$ may be required to be large as it is a non-adaptive Monte Carlo approximation in a high dimensional space. A variant is described where, every time a solution for $\mu$ is found, response function samples with low corresponding values of $\mu$ are replaced (again, from the given and non-adaptive Gaussian process). Although this now has the advantage of adapting the Monte Carlo samples to the problem, this has convergence problems that may be severe and not easy to diagnose.

In contrast, we formulate our adaptation of $\eta$ as a continuous optimization problem with an estimate of the gradient that has empirically reasonable stability, as expected from the related work in the machine learning literature for gradient estimation. We also parameterize the distribution so that the only constraint that we need to enforce concerns the univariate density $p(y \mid z)$ (or $p(y \mid x, z)$, in the variation discussed in Appendix G, which in principle requires no density estimation). Like the algorithm given by Gunsilius, the space of functions is a linear combination of a fixed dictionary of basis functions with a Gaussian distribution on the parameters, although we do not make use of the discrete mixture reweighting on the Monte Carlo samples, which introduces instability in (Gunsilius, 2020) despite its good theoretical properties. Our formulation, like the one in (Gunsilius, 2020), can in principle make use of more a flexible distribution such as a mixture of Gaussian copulas at the cost of more computation, as discussed in Appendix B. An important piece of future work is to thoroughly assess how stable a mixture of Gaussians version our algorithm is in practice.

The proposed implementation of Gunsilius' algorithm computes $F_{Y \mid do(x_0^\star)}(y^\star) - F_{Y \mid do(x_1^\star)}(y^\star)$, i.e., the difference in effects at two different treatment levels $x_0^\star$ and $x_1^\star$ for individuals within a fixed quantile $y^\star \in [0, 1]$ of the outcome variable. For example, in the expenditure dataset (see Section I.3), the setting $x_0^\star = 0.75, x_1^\star = 0.25, y^\star = 0.25$ would look at how much people, who spend a lot overall ($x^\star = 0.75$) and spend comparably little on food (up to 25%), would spend on food relatively to overall expenditure, if they spent much less overall ($x_1^\star = 0.25$). The main tuning parameter in the proposed algorithm is the penalization parameter $\lambda$, which corresponds to the tightness of the constraint. In the proposed implementation, this parameter is fixed throughout the optimization and must be chosen manually. In Figure 4, we show the results of Gunsilius's algorithm for three different levels of $y^\star$ on the expenditure dataset. Small values of $\lambda$ result in uninformatively loose bounds and do not always seem to converge (e.g., for $y^\star = 0.75$). As we increase $\lambda$, which corresponds to stronger enforcement of the constraint, the bounds get narrower. However, even after a long burn-in period, we still encounter substantial "instantaneous jumps" as well as longer-term drifts in the bounds, which may change the qualitative conclusions (for example in the $y^\star = 0.75$ setting). Note that this algorithm works on the empirical CDFs of all variables, i.e., they are all scaled to lie within $[0, 1]$.

Moreover, even after laboriously improving the performance of the algorithm using acceleration via JAX (Bradbury et al., 2018) and parallelized solving of the quadratic programs with CVXPY (Diamond & Boyd, 2016), producing an upper and lower bound for a single setting of $x_0^\star, x_1^\star, y^\star, \lambda$ with Gunsilius's algorithm took longer (about 30 minutes on a quad-core Intel Core i7) than a full set of upper and lower bounds at 15 different $x^\star$ values with our algorithm (about 20 minutes on the same hardware).

Figure 4: We show results of Gunsilius's algorithm for 3 different settings of $y^\star \in \{0.25, 0.5, 0.75\}$.

## B  The Shape of $p_\eta(\theta \,|\, x, z)$ and Conditional Effects

It is not difficult to show that our parameterization of $p_\eta(\theta \,|\, x, z)$ enforces $\theta \perp\!\!\!\perp Z$ while allowing for $\theta \not\!\perp\!\!\!\perp Z \,|\, X$, as suggested by Figure 1(c). It follows directly by factoring a conditional density in terms of a copula density $c(\cdot)$ and the required univariate marginals. That is, for some $(V_1, V_2, V_3)$ for which we want to define a conditional pdf $p(v_2 \,|\, v_1, v_3)$, we have

$$p(v_1, v_2 \,|\, v_3) := c(F(v_1 \,|\, v_3), F(v_2 \,|\, v_3))\, p(v_1 \,|\, v_3) p(v_2 \,|\, v_3) \quad \Rightarrow$$
$$p(v_2 \,|\, v_1, v_3) \,= c(F(v_1 \,|\, v_3), F(v_2 \,|\, v_3))\, p(v_2 \,|\, v_3).$$

Since $\int p(v_1, v_2 \,|\, v_3)\, dv_1 = p(v_2 \,|\, v_3)$, a necessary and sufficient condition for $V_2 \perp\!\!\!\perp V_3$ is choosing a model marginal such that $p(v_2 \,|\, v_3) = p(v_2)$. If $c(F(v_1 \,|\, v_3), F(v_2))$ cannot be factored in terms of some product $h_1(v_1, v_3) h_2(v_1, v_2)$, which is typically the case, then $V_2 \not\!\perp\!\!\!\perp V_3 \,|\, V_1$.

The main apparent limitation of our $p_\eta(\theta_k)$ (and the related copula) is its reliance on a parametric form. There is a complex relationship between the shape of the response function space and the distribution implied on that space by the unknown model $\mathcal{M}$. For $Y = f(X, U)$, it is always possible to assume without loss of generality that $U$ is a set of variables which are marginally standard Gaussians: just let the transformation $U'_i := \Phi^{-1}(F_i(U_i))$ be absorbed into $f(\cdot)$, where $F_i(\cdot)$ is the marginal CDF of $U_i$ and $\Phi(\cdot)$ is the CDF of a standard Gaussian. Moreover, assuming that any dependence among elements of $U$ can be explained by direct causation among them or by other latent parents, we can also assume all members of $U$ are independent.

However, we do not want to assume a one-to-one correspondence between elements of $\theta$ and elements of $U$: that is the whole point of using response functions. Even independent standard Gaussian $U$s would not translate to marginally Gaussian $\theta$. As an example, suppose $Y = U_1^2 X + \lambda U_2$. All response functions can be written in the form $f_\theta(x) := \theta_1 x + \theta_2$, where $\theta_1 = U_1^2$ and $\theta_2 = \lambda U_2$. Hence, $\theta_1$ follows a chi-squared distribution and $\theta_2$ a zero-mean, but not standard, Gaussian. If $Y = U_1 X^2 + \lambda U_1 U_2$, then on top of that $\theta_1$ and $\theta_2$ are not independent.

The solution is conceptually not complicated: just let $p_\eta(\cdot)$ be as flexible as desired. For instance, *let the copula be a finite or Dirichlet process mixture of Gaussian copulas, also defining flexible models for the marginals*. The IV conditional independence structure among $Z, X, \theta$ is still preserved. The practical issue of course is the optimization. The algorithm of Gunsilius (2020) itself tries to approach the problem by learning the reweighting of a Monte Carlo approximation to a fixed base measure. That alone is already very computationally demanding and has convergence problems.

We set a parametric form for $p_\eta(\cdot)$ for reasons beyond a compromise between flexibility and computational tractability. *Adopting a nonparametric model for the causal model, such as a Dirichlet process, seems pointless because:* (a) we do not perform statistical inference directly in the causal model, but only via black-box estimators of (features of) $p(x, y \,|\, z)$, which can be nonparametric; (b) if we were to follow the route of performing statistical inference by directly fitting the causal model, the corresponding estimator would have a finite representation with dimensionality given by the data. A practical resource, sample size, limits the representational size of the estimator. The role of nonparametrics is to provide a type of adaptive regularization, and to provide theory about limits of parametric estimators as done by Gunsilius (2020). The latter has clear value in itself but it does not demand nonparametric models to be actually implemented, while the former is out of our

scope: in our case, no regularization is needed for the causal model as we do not fit data based on it. Instead, our practical resource is the computational budget: if we want to not use domain knowledge to perform the causal analysis, we simply choose the size of $\eta$ based directly on the main bottleneck, the amount of computation available. Hence, by the time-data bounded nature of computational and statistical inference, we lose nothing by adopting a finite representation for both $\eta$ and $\theta$.

The practitioner should be invited to sample from the implied function space to visualize whether the distribution of sample paths has a desired level of variability. Getting the "exact" shape of the true distribution is however nowhere as important as just having enough variability to avoid overconfident bounds. How to achieve "enough variability" without aiming at a completely flexible distribution of $\theta$ may be a compromise between computational costs and domain-dependent judgment. In particular, given the choice of the $\{\phi_l\}$ family by which we link the causal model to observation, we may opt for the *maximum entropy distribution* that is given by the corresponding moments, the Gaussian in case of first and second moments of $p_\eta(\theta \mid x, z)$—although this still leaves open how the mean and covariance matrix of $\theta$ will change with $x$ and $z$.

In any case, the finite mixture of Gaussians approach can still be implemented with the reparameterization trick. The relation to Gunsilius algorithm is that our "base measure" is smoothly adaptive, leading to possibly more stable behavior in practice. The price to be paid is that each iteration in our method would be substantially more expensive than the efficient mixture component weighting optimization done at each iteration of Gunsilius' method, *if* we were to optimize the mixture component parameters to completion while fixing the samples. However, we do joint partial optimization by gradient-informed small steps, taken at each sampling stage. This is one of the main distinctive features of our class of algorithms compared to the resample/optimize alternating procedure of Gunsilius (2020).

To summarize, *the Gaussian case, discussed in the main text, should be seen as a useful illustration, not as a one-size-fits-all solution.* Any copula for which the reparameterization trick can be used can be automatically plugged into any instance of our class of algorithms.

Another important aspect brought by a parameterization of $p_\eta(\cdot)$ is in case we have pre-treatment covariates $W$ to either reduce confounding, remove (direct) dependence between $Z$ and $U$ or $Z$ and $Y$, or just to answer questions related to conditional expected outcomes, e.g., $\mathbb{E}[Y \mid do(x), w]$ and conditional average causal effects (CATE), $\mathbb{E}[Y \mid do(x), w] - \mathbb{E}[Y \mid do(x'), w]$. Although a response function can straightforwardly depend on a vector of treatment variables, this makes less sense if variables $W$ are not direct causes of $Y$. And even if elements of $W$ are direct causes, we may want to treat them analogously to $U$: playing a role in the response function only via the distribution of $\theta$, instead of being explicitly in the scope of such functions.

*Modeling CATE can then be done in a completely straightforward way.* Nothing in the algorithm changes if we use a probabilistic model for $p(x, y \mid z, w)$ to provide the observable counterpart of the causal model. Each configuration $w$ defines a separate optimization problem. The corresponding factor $p(\theta \mid x, z, w)$ can be set independently for each instance of $w$, regardless of its dimensionality.

However, a practitioner may be interested on providing information about how $p(\theta \mid x, z, w)$ varies smoothly across values of $w$ in order to impose further constraints on the response functions across multiple $w$ realizations. We suggest that a way of incorporating covariates $W$ is by a multilevel approach: define $p_{\eta(w)}(\theta \mid x, z, w)$, where each element of $\eta$ may itself be a function of $W$, e.g., $\mu_1 = \beta_1^\mathsf{T} W$ for some parameter vector $\beta_1$. Here, $p(x \mid z, w)$ and $p(y \mid z, w)$ (or $p(y \mid x, z, w)$) are the marginals to be matched. We will discuss in future work ways of making $p_\eta(\cdot)$ more flexible in general, including the use of covariates.

## C  Discrete Outcomes and Discrete Features

If $Y$ is discrete, $f_\theta(x)$ will be discontinuous. Theoretically this will not pose a problem as long as the number of discontinuities is finite (Gunsilius, 2020). The main practical issue is optimization, as eq. (6) will now not lead itself to gradient-based methods. The most immediate approximation is to use differentiable surrogates of $f_\theta(x)$ that relax the constraints. In the most basic formulation, we have the inequalities

$$tol_- \leq \mathbb{E}[\phi_l(Y) \mid z^{(m)}] - \int \phi_l(f_\theta(x)) \, p_\eta(x, \theta \mid z^{(m)}) \, dx \, d\theta \leq tol_+,$$

for some tolerance factors $tol_+, tol_-$. Given upper and lower bounds $\phi_l^+(f_\theta(x))$, $\phi_l^-(f_\theta(x))$ on $\phi_l(f_\theta(x))$, the relaxed constraints

$$tol_- \leq \mathbb{E}[\phi_l(Y) \mid z^{(m)}] - \int \phi_l^-(f_\theta(x)) \, p_\eta(x, \theta \mid z^{(m)}) \, dx \, d\theta$$

$$\mathbb{E}[\phi_l(Y) \mid z^{(m)}] - \int \phi_l^+(f_\theta(x)) \, p_\eta(x, \theta \mid z^{(m)}) \, dx \, d\theta \leq tol_+,$$

will still result in valid, but looser bounds (again, up to local optima and Monte Carlo error). If $f_\theta(x)$ is non-negative (for instance, if its codomain is $\{0, 1\}$) and $\phi_l(\cdot)$ is monotonic for non-negative inputs (such as $\phi_l(x) = x$ and $\phi_l(x) = x^2$), it is enough to plug in bounds for $f_\theta(x)$ itself. We will elaborate on that in future work. In this context, we can also formulate an alternative approach to matching $p(y \mid z)$.

**Alternative Approach to Matching** $p(y \mid z)$**.** Here we describe an alternative approximation of eq. (5) that hinges on smoothly approximating the indicator function to render the integral well behaved. First, instead of evaluating $\Pr(Y < y \mid Z = z^{(m)})$ for all $y \in \mathcal{Y}$, we take a similar approach for discretizing $Y \mid z^{(i)}$ as we took for $z^{(m)}$. For a given $z^{(m)}$, instead of all half-spaces $Y < y$, we only consider the sets

$$A^{(m,l)} := (-\infty, y^{(m,l)}] \quad \text{with} \quad y^{(m,l)} := F_{Y \mid z^{(m)}}^{-1}\left(\frac{l-1}{L-1}\right)$$

for $l \in [L]$ with some fixed $L \in \mathbb{N}$. This results in constraints for the $L$-quantiles of the conditional distributions of $Y$

$$\frac{l-1}{L-1} = \int \mathbf{1}\left(f_\theta(x) \leq y^{(m,l)}\right) \, p_\eta(x, \theta \mid z^{(m)}) \, dx \, d\theta.$$

for all $m \in [M]$ and $l \in [L]$. In practice, we would evaluate the integral on the right hand side with a Monte Carlo estimate, sampling from $p_\eta(x, \theta \mid z^{(m)})$ and then differentiate with respect to $\eta$ for gradient-based optimization. Therefore, the non-differentiable (even non-continuous) indicator function poses an issue for the optimization. We can circumvent this problem by approximating the indicator with a smoothly differentiable function, for example

$$\mathbf{1}(t \leq t^*) \approx \sigma_\rho(t - t^*) \quad \text{for} \quad \sigma_\rho(t) := \frac{1}{1 + e^{-\rho t}} \quad \text{or} \quad \sigma_\rho(t) := \frac{1}{1 + \exp\left(-\rho\left(t + \frac{1}{\sqrt{\rho}}\right)\right)}$$

for $\rho > 0$. As $\rho \to \infty$, $\sigma_\rho(t) \to \mathbf{1}(t \leq 0)$ pointwise on $\mathbb{R} \setminus \{0\}$, i.e., we can slowly increase $\rho$ throughout the optimization to gradually approximate the constraints.

Hence an alternative approach to implement the constraint for matching $p(y \mid z)$ is

$$\frac{l-1}{L-1} = \int \sigma_\rho\left(f_\theta(x) - y^{(m,l)}\right) \, p_\eta(x, \theta \mid z^{(m)}) \, dx \, d\theta$$

for all $m \in [M]$ and $l \in [L]$, where we increase $\rho > 0$ after each optimization round.

In practice, we this approach gave less robust results than the approach described in the main text, partly due to the additional hyperparameter schedule needed for $\rho$. Therefore, we only report results for the approach using dictionary functions $\phi_l$ described in the main text.

# D  Algorithm

## D.1  Additional Details of the Optimization

**Smoothen** LHS**.** Since $\text{LHS}_{m,l}$ are estimated via empirical averages of $\phi_l(y_i)$ for datapoints in a given bin $i \in \text{bin}^{-1}(m)$, "neighboring" constraints $\text{LHS}_{m,l}$ and $\text{LHS}_{m+1,l}$ may have substantially different values. Since our model is smooth, it can be hard to match such non-continuities with $\text{RHS}_{m,l}(\eta)$. Intuitively, we expect such jumps to be artifacts of finite sample effects and not important properties of the true data distribution. Hence we apply a spline regression to the values $\{\text{LHS}_{m,l}\}_{m=1}^M$ for each $l \in [L]$ to smoothen out larger jumps between neighboring values. In practice, we use a cubic univariate spline for each $l$ with a smoothing factor of $0.2$.

---

**Algorithm 1** Bounding the IV interventional effect at treatment level $x^\star$.

---

**Require:** dataset $\mathcal{D} = \{(z_i, x_i, y_i)\}_{i=1}^N$; number of $z$ grid points $M$; constraint functions $\{\phi_l\}_{l=1}^L$; response function family $\{f_\theta\}_{\theta \in \Theta}$; batchsize $B$; initial temperature $\tau^{(0)} > 0$; temperature increase factor $\alpha > 1$; tolerances $\epsilon_{\text{abs}}, \epsilon_{\text{rel}}$; initial Lagrange multipliers $\lambda$; initial parameters $\eta^{(0)}$;

1: $z^{(m)} := \hat{F}_Z^{-1}(\frac{m}{M+1})$ for $m \in [M]$ &emsp;&emsp;&emsp;&emsp;&emsp;&emsp; $\triangleright \hat{F}_Z$: CDF of $\{z_i\}_{i=1}^N$.

2: $\text{bin}(i) := \max\{\arg\min_{m \in [M]} |z_i - z^{(m)}|\}$ for $i \in [N]$ &emsp;&emsp; $\triangleright$ split data points into "z-bins"

3: $\text{LHS}_{m,l} := \frac{1}{|\text{bin}^{-1}(m)|} \sum_{i \in \text{bin}^{-1}(m)} \phi_l(y_i)$ for $m \in [M], l \in [L]$ &emsp;&emsp; $\triangleright$ pre-compute LHS

4: smoothen $\text{LHS}_{m,l}$ across $m$ for each $l$ with spline regression &emsp;&emsp;&emsp; $\triangleright$ see Appendix D.1

5: $b := \max\{\epsilon_{\text{abs}}, \epsilon_{\text{rel}} \text{ LHS}\}$ (element-wise) &emsp;&emsp;&emsp;&emsp; $\triangleright$ set constraint tolerances

6: $\hat{x}_j^{(m)} := \hat{F}_{X \mid z^{(m)}}^{-1}\left(\frac{j-1}{B-1}\right)$ for all $j \in [B], m \in [M]$ &emsp;&emsp; $\triangleright \hat{F}_{X \mid z^{(m)}}$: CDF of $\{x_i\}_{i \in \text{bin}^{-1}(m)}$

7: **for** $t = 1 \ldots T$ (or until convergence) **do** &emsp;&emsp;&emsp;&emsp;&emsp; $\triangleright$ optimization rounds

8: &emsp;&emsp; $\eta^{(t)} := \text{OPTIMIZESUBPROBLEM}(\eta^{(t-1)}, \lambda^{(t-1)}, \tau^{(t-1)})$ &emsp;&emsp; $\triangleright$ min. Lagrangian at fixed $\lambda, \tau$

9: &emsp;&emsp; $\lambda_l^{(t)} \leftarrow \max\left(0, \lambda_l^{(t-1)} - \tau^{(t-1)} c_l(\eta^{(t)})\right)$ &emsp;&emsp; $\triangleright$ update Lagrangian multipliers

10: &emsp;&emsp; $\tau^{(t)} \leftarrow \alpha \, \tau^{(t-1)}$ &emsp;&emsp;&emsp;&emsp;&emsp;&emsp; $\triangleright$ increase temperature parameter

11: **return** $o_{x^\star}(\eta^{(T)})$

12: **function** OPTIMIZESUBPROBLEM$(\eta, \lambda, \tau)$

13: &emsp;&emsp; $\triangleright$ In here we use SGD with auto-differentiation to minimize $\mathcal{L}$. Hence we only describe how to evaluate $\mathcal{L}$ in a differentiable fashion:

14: &emsp;&emsp; $o_{x^\star}(\eta) := \frac{1}{B} \sum_{j=1}^B f_{\theta^{(j)}}(x^\star)$ with $\theta^{(j)} \sim p_\eta(\theta)$ &emsp;&emsp; $\triangleright$ c.f. Algorithm 2 for sampling

15: &emsp;&emsp; $\text{RHS}_{m,l}(\eta) := \frac{1}{B} \sum_{j=1}^B \phi_l\left(f_{\theta^{(j)}}(\hat{x}_j^{(m)})\right)$ &emsp;&emsp; $\triangleright$ c.f. Algorithm 2 for sampling

16: &emsp;&emsp; $c(\eta) := b - |\text{LHS} - \text{RHS}(\eta)|$ &emsp;&emsp;&emsp;&emsp; $\triangleright$ compute constraint terms

17: &emsp;&emsp; $\mathcal{L}(\eta) := \pm o_{x^\star}(\eta) + \sum_{l=1}^{M \cdot L} \xi(c_l(\eta), \lambda_l, \tau)$ &emsp; $\triangleright$ Lagrangian ($\pm$ for lower/upper bound)

18: &emsp;&emsp; **return** $\arg\min_\eta \mathcal{L}(\eta)$ &emsp;&emsp;&emsp;&emsp;&emsp; $\triangleright$ optimize with SGD

---

## D.2 &emsp; Augmented Lagrangian Optimization Strategy

The Augmented Lagrangian method (Hestenes, 1969) is a general method for constrained optimization, originally proposed just for dealing with equality constraints. The benefit of this over penalty methods is that we do not need to take the penalty parameters $\tau$ to $\infty$ in order to solve the original constrained optimization problem, which can cause ill-conditioning (Nocedal & Wright, 2006). However, our problem only contains inequality constraints. Thus, we consider a refinement proposed by Nocedal & Wright (2006) to purely handle inequality constraints using Augmented Lagrangian methods. Specifically, we can write the inequality constrained optimization problem equivalently as an unconstrained optimization problem with Lagrange multipliers $\lambda$:

$$\min_\eta \max_{\lambda \geq 0} \left\{ o_{x^\star}(\eta) + \lambda^\top (c(\eta) - b) \right\}.$$

To see that it is equivalent, note that the $\max$ returns $o(\eta)$ when $\eta$ satisfies the constraints (as the maximum is obtained at $\lambda = 0$), and $\infty$ otherwise (as the maximum is at $\lambda = \infty$). However, this is not easy to optimize as the $\lambda$ jumps from 0 to $\infty$ when passing through the constraint boundary. To fix this, we add a term that penalizes $\lambda$ making larger changes from its previous value. Specifically,

$$\min_\eta \max_{\lambda \geq 0} \left\{ o(\eta) + \lambda^\top (c(\eta) - b) - \frac{1}{2\tau} \|\lambda - \lambda'\|^2 \right\},$$

where $\lambda'$ are the Lagrange multipliers from the previous iteration and $\tau$ is a penalty term that is iteratively increased. Note that the $\max$ optimization can be solved in closed form for each Lagrange multiplier $\lambda_l$

$$\lambda_l = \max\left\{0, \lambda_l' + \tau c_l(\eta)\right\},$$

---

**Algorithm 2** Sampling parameter values $\theta$ from $p_\eta(\theta, X \mid z^{(m)})$.

---

1: Sample each component of $w \in \mathbb{R}^{K \times B}$ i.i.d. from a standard Gaussian.

2: Prepend the vector $(0, 1/B, \dots, 1)$ as the first row of $w$, resulting in $w \in \mathbb{R}^{(K+1) \times N}$.

3: Allow for dependencies between components by multiplying with the Cholesky factor $w \leftarrow L\,w$.

4: Normalize all values by applying the standard Gaussian CDF component wise, $w \leftarrow \varphi_{0,1}(w)$.

5: Fix the marginals of $\theta_k^{(j)}$ by applying the inverse CDF of a $(\mu_k, \sigma_k^2)$-Gaussian: $\theta^{(j)} \leftarrow \varphi_{\mu_k, \sigma_k^2}^{-1}(w_{j+1})$ for $j \in [K]$. Here, $w_{j+1}$ denotes the $j+1$-st row of $w$.

6: Sampling $X$ via $\hat{F}_X^{-1}(w_1)$ by design simply gives the pre-computed $\hat{x}$.

---

where $c_l(\eta)$ is shorthand for the $l$-th inequality constraint. Plugging these values into the optimization problem, we arrive at

$$\min_\eta \mathcal{L}(\eta, \lambda, \tau) := o_{x^\star}(\eta) + \sum_{l=1}^{M \cdot L} \xi(c_l(\eta), \lambda_l, \tau)$$

with

$$\xi(c_l(\eta), \lambda_l, \tau) := \begin{cases} -\lambda_l c_l(\eta) + \frac{\tau c_l(\eta)^2}{2} & \text{if } \tau c_l(\eta) \le \lambda_l, \\ -\frac{\lambda_l^2}{2\tau} & \text{otherwise,} \end{cases}$$

where $\tau$ is increases throughout the optimization procedure. Given an approximate solution $\eta$ of this subproblem, we then update $\lambda$ according to

$$\lambda_l \leftarrow \max\{0, \lambda_l - \tau c_l(\eta)\}$$

for all $l \in [M \cdot L]$ and set $\tau \leftarrow \alpha\tau$ for a fixed $\alpha > 1$. For the full optimization, we attach temporal upper indices, i.e., at time step $t$, we have the current approximate solution $\eta^{(t)}$, the Lagrange multipliers $\lambda_l^{(t)}$ and the temperature parameter $\tau^{(t)}$. See Algorithm D for a description of the optimization scheme. While the number of optimization parameters grows quickly with the dimensionality of $\theta$, which may render the optimization challenging, in our experiments we did not encounter any issues with up to 54 optimization parameters and 40 constraints.

### D.3 Sampling from the Copula

A crucial step for our algorithms was baking the assumptions about $p(X \mid Z)$ as well as $Z \perp\!\!\!\perp U$ directly into our model from which we sample for Monte Carlo estimates. Algorithm 2 describes in detail how we can obtain these samples from the copula defined in eq. (4) in a differentiable fashion with respect to $\eta$.

### D.4 Parameter Initialization

We initialize the optimization parameters $L$ with ones on the diagonal, zeros in the upper triangle, and sample the lower triangle from $\mathcal{N}(0, 0.05)$. The initialization for $\mu_k$ and $\ln(\sigma_k^2)$ depends on the chosen response function family. Our guiding principle is to ensure that the initial distribution covers a large set of possible response functions, tending towards larger $\sigma_k$.

## E   Response Functions

One key advantage of our approach is that it allows us to flexibly trade off assumptions on the response function family with more informative bounds. Due to our simple, yet expressive choice of linear combinations of a set of basis functions, there are many natural and easy to implement options for the response functions. In particular, we consider the following options:

1. *Polynomials:* $\psi_k(x) = x^{k-1}$ for $k \in [K]$. In this work, we specifically focus on linear ($K = 2$), quadratic ($K = 3$), and cubic ($K = 4$) polynomial functions.

2. *Neural basis functions (MLP):* We fit a multi-layer perceptron with $K$ neurons in the last hidden layer to the observed data $\{(x_i, y_i)\}_{i \in N}$ and take $\psi_k(\text{x})$ to be the activation of the $k$-th neuron in the last hidden layer. Note that the network output itself is a linear combination of these last hidden layer activations. Hence, the underlying assumption for this approach to work well is that the true causal effects can also be approximated well by a linear combination of the learned last hidden layer activations, i.e., the true effect is in this sense "similar" to the estimated observed conditional $\hat{p}(y \,|\, x)$. In practice, we train a 2-hidden layer MLP with 64 neurons in each layer, rectified linear units as activation functions and an mean-squared-error loss for 100 epochs and a batchsize of 256 using Adam with a learning rate of 0.001.

3. *Gaussian process basis functions (GP):* We fit a Gaussian process with a sum-kernel of a polynomial kernel of degree 3, an RBF kernel, and a white noise kernel to $K$ different sub-samples $\{(x_i, y_i)\}_{i \in N'}$ with $N' \leq N$. We then sample a single function from each Gaussian process as the basis functions $\psi_k$ for $k \in [K]$. We train multiple Gaussian processes on smaller subsets of the data to ensure sufficient variance in the learned functional relation. Similarly to the neural net basis functions, the assumption is that the causal effect can be approximated by a linear combination of these varying samples. In our experiments, we fit the Gaussian processes with scikit-learn's `GaussianProcessRegressor` (Pedregosa et al., 2011) using $N' = 200$ and a white kernel variance of 0.4.

# F    Why Discretization is not a Good Idea

The framework of Balke & Pearl (1994) is powerful and simple, and hence it raises the prospect that discretizing treatment $X$ can provide a good approximation to the original problem where $X$ is continuous. However, there are several reasons why this is not a good idea:

- *It destroys the key assumption of instrumental variable modeling.* Besides the lack of confounding between instrument $Z$ and outcome $Y$, the key assumption in an IV model is the conditional independence $Y \perp\!\!\!\perp Z \,|\, \{X, U\}$ ("exclusion restriction"). This assumption will in general fail to hold if we destroy information, i.e., if we condition on $X \in \mathcal{A}$, for some set $\mathcal{A}$, instead of the realization of $X$;

- *It makes causal estimands ill-defined.* There are several ways in which an intervention can be ambiguous. This happens when defining the manipulation of a construct ("race") or of summary measurements in general ("obesity"). One particular instance of the latter is when we speak of $do(x^\star)$, meaning the setting of a discretization $X^\star$ of $X$ to a particular level $x^\star$ (VanderWeele & Hernán, 2013). If $X^\star = x^\star$ corresponds to the event $X \in [a, b]$, then this at least needs the assumption that $\mathbb{E}[Y \,|\, do(x)]$ is approximately constant for $x \in [a, b]$ for the intervention to be meaningful. This is pointless if the goal is to avoid making assumptions about the shape of the response function;

- *Its cost is super-exponential.* Suppose we still want to proceed with the idea of discretization, in the sense that we are willing to assume that we are using a fine enough grid of intervals for the treatment so that the previous two points are not particularly prominent. It may be argued that using Balke & Pearl (1994) with this approximation is attractive on the grounds it is a convex, deterministic approach and hence a more computationally attractive alternative to tackling the continuous problem. In fact, the opposite may hold. Assume we discretize $X$ and $Y$ to $|\mathcal{X}|$ and $|\mathcal{Y}|$ levels respectively, and $Z$ assumes $|\mathcal{Z}|$ levels (perhaps also by discretization). Then the cost of using the full information of the distribution is approximately $\mathcal{O}(|\mathcal{X}|^{|\mathcal{Z}|} |\mathcal{Y}|^{|\mathcal{X}|})$. It is true that, just like in our approach, this can be much simplified if we rely only on a subset of constraints. In particular, if we use only the first moments in the constraints and the expected outcome is the objective function, we can simplify the discrete formulation by targeting our parameterization to depend only on the expected outcomes directly. This makes the problem exponential only on $|\mathcal{Z}|$, see for instance the parameterization of Zhang & Bareinboim (2020). Being "only" exponential may still require Monte Carlo approximations in general. But this can still be super-exponential if $|\mathcal{Z}|$ grows with $|\mathcal{X}|$, which will be necessary if the instrument is strong: for an extreme example, if $Z$ and $X$ lie close to a line with high probability and we choose only two levels of $Z$ against many levels of $X$, then most combinations of pre-determined $(z, x)$ pairs will lie on regions of essentially zero density in the $p(x, z)$ distribution;

- *It is vacuous in the limit.* Even if we can use an arbitrarily fine discretization and assume that the piecewise nature of the approximation is close enough to the true response functions of $Y$, we know that as $|\mathcal{X}| \to \infty$ the number of discontinuities in the response function also goes to infinity.

As described by Gunsilius (2018), we will not learn anything non-trivial about the causal estimand of interest.

We reiterate the points above in more direct way: *being unable to express constraints on the response function is not an asset, it's a liability.* Discretization allows us to easily use a single family of functional constraints: piecewise constant functions. In this framework, it is cumbersome to represent other constraints such as smoothness constraints, and *the degree of violation of the exclusion restriction assumption remains unknown.* There is no reason to believe this discrete representation is a good family in any computationally bounded sense, as an efficient choice of discretization points can only be made if we know something about the function. And if we do, then it makes far more sense to use more representationally efficient ways of partitioning the space of $X$, such as regression splines with a fixed number of knots. This involves no discretization of treatment, while avoiding the issues of violation of the exclusion restriction assumption and ambiguity of intervention.

## G  Modeling $p(y \mid x, z)$ and Monte Carlo Alternatives

Alternatively to the setup described in the main text, we can match not only the marginal $p(y \mid z)$, but the theoretically more informative $p(y \mid x, z)$. This problem is actually conceptually simpler, although it will require joint measurements over the three types of variables.

The main modification is as follows. Instead of

$$\mathbb{E}[\phi_l(Y) \mid Z = z^{(m)}] = \int \phi_l(y_\theta(x)) \, p_\eta(x, \theta \mid Z = z^{(m)}) \, dx \, d\theta,$$

we build constraints based on

$$\mathbb{E}[\phi_l(Y) \mid X = x^{(m)}, Z = z^{(m)}] = \int \phi_l(y_\theta(x^{(m)})) \, p_\eta(\theta \mid X = x^{(m)}, Z = z^{(m)}) \, d\theta,$$

where now we need to define a grid over the joint space of $X$ and $Z$. This can be done in several ways, including the joint product of equally-spaced quantiles of the respective marginal distributions, perhaps discarding combinations for which $p(x^{(m)}, z^{(m)})$ are below some threshold. Moreover, the factor $p_\eta(\theta \mid X = x^{(m)}, Z = z^{(m)})$ was explicitly parameterized in our original setup, and can be used as is.

Notice the advantages and disadvantages of the two approaches. Modeling the full conditional $p(y \mid x, z)$ uses the full information of the problem (as it is equivalent to $p(x, y \mid z)$, where $p(x \mid z)$ is tackled directly), which in principle is more informative but requires functionals of the joint $p(x, y \mid z)$ instead of the marginals $p(x \mid z)$ and $p(y \mid z)$. We can also see that we are trading-off adding more constraints but removing the need to integrate $X$ in each constraint. More interestingly, this full conditional approach does not require any kind of density estimation: the need for $p(x \mid z)$ disappears, and all we need on the left-hand sides are estimates of expectations.

More generally, it is clear that there are practical cases where $\mathbb{E}[\phi_l(y_\theta(X)) \mid x, z]$, with $\theta$ being the random variable to be marginalized, has an analytical solution as a function of $\eta$. For instance, this will be the case when $\phi_l$ stands for linear and quadratic functions of $Y$ (itself a linear function of $\theta$ for a fixed $x$), and $p_\eta(\theta \mid x, z)$ is a (mixture of) Gaussian(s), which is true for our experiments. However, we demonstrate the suitability of Monte Carlo formulation in order to provide and evaluate a class of algorithms for black-box (differentiable) features, where such expectations cannot be computed analytically in general.

## H  Fitting Latent Variable Models

When fitting the latent variable models, we use multi-layer perceptrons with inputs $z, x, y$ for the means and variances of the latent dimensions $U$, where we use lower indices $U_i$ for the different components. For this encoder, we use 32 neurons in the hidden layer and rectified linear units as the activation function. There are two decoders. The first one is trained to reconstruct $\mathbb{E}[X \mid X, U]$, i.e., receives the original $Z$ in addition to the latent vector $U$ as input. It is also parameterized by an MLP with 32 neurons in the hidden layer and ReLu activations. The second decoder reconstructs $\mathbb{E}[Y \mid X, U]$ and is either an MLP of the same architecture (when comparing to MLP response functions), linear in $X$, i.e., $\alpha X + \beta + \sum_{i=1}^{\text{n\_latent}} (\gamma_i X U_i + \delta_i U_i)$ (when comparing to linear response

Figure 5: Bounds for the simulated sigmoidal design. The true causal effect is given by a logistic function, which is well recovered by our method for different response function families (cubic polynomials, GP basis functions, and MLP basis functions).

functions), or quadratic in $X$, i.e., $\alpha X^2 + \beta X + \gamma + \sum_{i=1}^{\text{n\_latent}} (\delta_i X^2 U_i + \epsilon_i X U_i + \zeta_i U_i)$ (when comparing to quadratic response functions). Thereby, we ensure that the form of matches our assumptions on the function form of the response family. We then optimize the evidence lower bound following standard techniques of variational autoencoders (Kingma & Welling, 2014) with $L_2$ reconstruction loss for $X$ and $Y$. We fit multiple models with different random initializations and compute the implied causal effect of $X$ on $Y$ for each one, which is obtained from the decoder $\mathbb{E}[Y \mid u, x]$ by averaging over 1000 samples of the latent variable $U$ for a fixed grid of $x$-values.

# I  Additional Experimental Results

## I.1  Hyperparameter Settings

In all experiments, we fix hyperparameters $M = 20$, $L = 2$, $B = 1024$ and run SGD with momentum 0.9 and learning rate 0.001 for 150 rounds of the augmented Lagrangian with 30 gradient updates for each subproblem optimization. We start with a temperature parameter $\tau = 0.1$ and multiply it by $\alpha = 1.08$ in each round, capped at $\tau_{\max} = 10$. We use 7 neurons in the last hidden layer of the feed-forward neural net for MLP response functions in our synthetic setting and 9 for the expenditure data. For GP basis functions (see Appendix E), we sample 7 basis functions for the sigmoidal design dataset (see Appendix I.2). This set of hyperparameters did not require much manual tuning and worked for all datasets and response function families, i.e., also different dimensionality of $\theta$. For the synthetic settings, we sample 5000 observations each. We use 3 as the latent dimension when fitting our latent variable models. For the tolerances, we use $\epsilon_{\text{abs}} = 0.2$ for the synthetic settings, $\epsilon_{\text{abs}} = 0.1$ for the sigmoidal design (see Section I.2), $\epsilon_{\text{abs}} = 0.3$ for the expenditure dataset and gradually tighten $\epsilon_{\text{rel}}$ from 0.3 to 0.05 in all settings (which corresponds to the increasingly opaque lines).

## I.2  Sigmoidal Design

We also evaluate our method on simulated data from a sigmoidal design introduced by Chen & Christensen (2018), adopted by Newey & Powell (2003) and used in previous work on continuous instrumental variable approaches under the additive assumption as a common test case (Hartford et al., 2017; Singh et al., 2019; Muandet et al., 2020). We show the results from KIV and our bounds for response function families consisting of cubic polynomials and neural net basis functions in Figure 5. The observed data distribution $\hat{p}(y \mid x)$ follows the true causal effect rather closely and the instrument is relatively strong in this setting, see Singh et al. (2019) for details. Therefore, the gap between our bounds is relatively narrow for a broad set of different basis functions as long as they are flexible enough to capture a sigmoidal shape.

## I.3  Expenditure Data

We prepare the data from Office for National Statistics (2000) using the same steps as Gunsilius (2020) closely following Newey & Powell (2003); Blundell et al. (2007). This is, we restrict the sample to households with married couples who live together and in which the head of the household is between 20 and 55 years old. We further exclude couples with more than 2 children. Finally,

Figure 6: Performance of our method on smaller datasets with only 500 observations. The left column is the strong confounding weak instrument case ($\alpha = 0.5, \beta = 3$) and the right column is the weak confounding strong instrument case ($\alpha = 3, \beta = 0.5$).

Figure 7: Results for the small dataset from Acemoglu et al. (2001) with linear response functions and $M = 5$ $z$-bins.

we also require the head of the household not to be unemployed. Otherwise, the instrument, gross earnings, would not be available. After these restrictions, we end up with 1650 observations in our dataset. The dataset can be downloaded for free for academic purposes after creating an account.

## I.4 Small Data Regime

Having tested our method on datasets of size 5000 (synthetic) and 1650 (expenditure data, see Appendix I.3), we now evaluate how our method performs on even smaller datasets. To this end, we first look at our synthetic settings using only 500 datapoints and correspondingly reducing the number of $z$-bins to $M = 6$ in Figure 6. While the bounds are looser, our method can still provide useful information with relatively little data.

In addition, we ran our methods on a classic instrumental variable setting from economics, namely the dataset used by Acemoglu et al. (2001) on using settler mortality as an instrument to estimate

linear Gaussian setting with strong confounding and weak instrument ($\alpha = 0.5$, $\beta = 3$)

non-linear, non-additive setting with strong confounding and weak instrument ($\alpha = 0.5$, $\beta = 3$)

Figure 8: We show the results of a manual hyperparameter search for KIV in the left column, where we score different settings in the two-dimensional hyperparameter space by the log of the out-of-sample mean squared error, which requires knowledge of the true causal effect. The red cross denotes the setting with the smallest out-of-sample mean squared error. In the right column, we show the KIV regression lines using the hyperparameters found in the manual search. The first row corresponds to the linear Gaussian setting and the second row to the non-linear, non-additive synthetic setting.

the causal effect of the health of institutions on economic performance.[10] This dataset consists of only 70 datapoints. Therefore, we set the number of $z$-bins to $M = 5$ for this dataset. Restricting ourselves to linear response functions, our method still gives informative bounds, which include the effect estimated by 2SLS, but does not fully include the KIV results, see Figure 7.

## J  KIV Heuristic for Tuning Hyperparameters

We have found KIV to fail in the strongly confounded linear Gaussian setting, even though all the assumptions are satisfied, see Figure 2 (row 1). Closer analysis of these cases showed that the heuristic that determines the hyperparameters does not return useful values in this setting. Instead, we performed a grid search over the main hyperparameters $\lambda$ and $\xi$ (see Singh et al., 2019, for details) and scored them by the out-of-sample mean-squared-error for the true causal effect (which is known in our synthetic setting). After manual exploration of the parameter space, we found a good setting marked by the red cross in the first row on the left of Figure 8. Using these fixed hyperparameters for KIV instead of the internal tuning stage, we get a much better approximation of the true causal effect shown in the first row on the right of Figure 8. Towards the data starved regions at large and small $x$-values, KIV again reverts back towards the prior mean of zero as expected. It is unclear at the moment, however, how to set such hyperparameter values without access to the true causal effect. Our point here is that in principle there is a setting with acceptable results, although even then it is not clear how much of it is a coincidence based on looking at many possible configurations.

We performed a similar manual analysis for the non-linear, non-additive synthetic setting with strong confounding, in which off-the-shelf KIV fails as well, see Figure 2 (row 3). Note that this setting does not satisfy the assumptions of KIV, because of the non-additive confounding. Again, we do manage to find hyperparameters that locally minimize the out-of-sample mean-squared-error shown

in the second row on the left of Figure 8. However, the resulting regression of the causal effect does not properly capture the true effect as shown in the second row on the right of Figure 8.

## Footnotes

[10]The dataset is freely available at `https://economics.mit.edu/faculty/acemoglu/data/ajr2001`.