[Reviews · NeurIPS 2020]

Review 1

Summary and Contributions: The authors study the problem of partial identification of causal effect, which provides bounds on the causal effect from a treatment variable on an outcome variable. They consider the case that there exists an observable instrumental variable and the treatment and outcome are continuous variables. In the proposed method, a parametric response function family is considered. Then the causal effect, written as an integral over the distribution of possible responses to the treatment is approximated. Hence, efficient gradient-based optimization techniques can be used to find lower/upper bounds on the causal effect.

Strengths: Majority of the work on instrumental variable framework consider additive noise models to enable identification, and works on partial identification usually do not consider continuous distributions. Hence this work provides an efficient method for the missing case of partial identification of continuous models. This makes the work interesting to the community of causal inference. The approach for optimization seems sound and interesting.

Weaknesses: - Since the proposed optimization is non-convex, there is no guarantee for the correctness of the bounds. - Perhaps the most important missing result in this work is confidence intervals for the bounds. - At some parts, for instance the choice of function family for p_\eta, it seems that the only criteria for the choices in the model is to make the optimization task efficient and no other justification is provided. - Is there any intuitions or guidelines for choosing the response functions? I thought MLP should be a good choice, but the resulting bounds seem to be loose. - In the experiments, only two cases are considered: linear Gaussian case and a second case in which the treatment is again linear and the outcome is generated by 0.3X^2−1.5XC+e. It seems necessary to consider other instances of non-linear cases as well. - The choice of the outcome equations (X-6C+e and 0.3X^2−1.5XC+e) look random. Was there any specific reason for this choice?

Correctness: The method used in this work seems correct to me.

Clarity: Overall the paper is easy to read, yet some details such as details of subsection 3.2 could have been explained more.

Relation to Prior Work: Yes.

Reproducibility: Yes

Additional Feedback: After rebuttal: I thank the authors for their responses. My score remains the same.


Review 2

Summary and Contributions: The paper presents a response function [Balke & Pearl] approach for bounding the causal effect in instrumental variable settings with continuous treatment and response variables. Their approach optimizes over a parametric response function space via an augmented legrangian procedure (to deal with the constrained optimization).

Strengths: Overall I enjoyed this paper - it demonstrates how we can leverage modern ideas such as the reparameterization trick to address the computational challenges associated with bounding the causal effect in continuous settings. Perhaps more importantly, it complements the ever-expanding toolkit of machine learning methods for IV to include a partial identification method: addressing uncertainty from potential lack of identification is an important topic for practical causal inference.

Weaknesses: My biggest concern is the sensitivity of the method to parametric assumptions. This is obviously unavoidable --- as you grow the space of possible models, you also worsen the identification problem --- but I would have liked to see some discussion of the tradeoffs here. The paper points out the limitations of the y = f(x) + e_y approach to achieving identification; but then isn't as explicit about the implied limitations of different parametric assumptions. Section G in the appendix deals with some of this, but I still have a hard time thinking about how an analyst might reason about the tradeoffs of different parameterizations of the models and their associated assumptions about the ways that u can affect f in the structural equation.

Correctness: As far as I can tell - yes. Both the methods and the empirical evaluation appear sound.

Clarity: The paper is very clear.

Relation to Prior Work: Most of the relevant work is cited, but I would include Bennet, Kallus & Schnabel [2019] & Lewis & Syrgkanis [2018] among the deep network approaches.

Reproducibility: Yes

Additional Feedback:


Review 3

Summary and Contributions: The authors are interested in deriving upper / lower bounds for causal effects under the assumption of existence of an instrumental variable, by maximizing / minimizing the causal effect estimation over all IV models compatible with the observed data distribution. They propose to build on work from [Balke&Pearl (1994)] - who describe constraints for the compatibility of the model with the observed data distribution in a sharper way than most recent works - and solve the untractable optimisation problem that arises using recent advances in SGD/Monte-Carlo methods.

Strengths: The article is clearly written and the authors are very pedagogic in explaining their contributions, and how they relate to prior work. The authors notably build on the pioneering work from Balke and Pearl to define the constraints that define the “models compatible with the observed data” using marginals, making a clear difference with most recent work on this matter. The proposed method has two main advantages: (1) it deals with continuous treatment and (2) allows for the use of recent stochastic optimisation algorithms - both of which are, as far as I know, novel. This is done by parametrising the causal function space in a simple (yet expressive) way that allows to incorporate constraints for the underlying model. The experiments are very thorough. The authors experiment both with a linear additive case and non-linear non-additive case, each time with varying levels of confounding, and instrument strength. They compare their bounds with the true causal effect, but also report results from 2SLS and the recent KIV method.

Weaknesses: The authors make choices regarding the parametrisation of the various distributions at play, which are consistent with experimentation and implementation choices. Although this is understandable in the case of such a complex problem, some minor comments/questions remain. The response function space is modelled as the space of linear combination of basis functions. While the authors argue that the proposed method works for any differentiable parametrisation, it isn’t clear how the optimisation algorithm would behave if the response function space was not parametrised as such. Such a parametrisation is indeed very expressive (as explained in appendix), and is valid if we have prior knowledge of the response function form (which seems to be what is assumed in the experiments in line 246). However, one may wonder how to choose such basis functions in practice, when one has no prior knowledge on the confounders, which might have any type of complex influence on the other variables at play. In such a case, the proposed parametrisation could lead to overlooking part of the (valid) response function space, possibly invalidating the optimisation result. The authors propose to build a grid for variable Z, enabling simpler matching of p(y|z) notably. As mentioned in the article, such an “approximation can only relax the constraints”, and therefore not invalidate the bounds, although the extent to which the bounds might be loosened isn’t clearly discussed. In Section 3.2, the authors propose to “bake in” one of the constraint on the marginal p(x|z) by directly using the p(x|z) identified from the observed data (line 154), although it isn’t clear how this is done exactly. As far as I understand, in the experiments the authors refer to each point z of the Z grid, and consider corresponding observations of X to estimate p(X|Z=z): wouldn’t cases where there are few values of X for a given value z be problematic ? Wouldn’t identifying such a distribution from the data imply to have a prior model for this distribution (e.g. linear Gaussian) ?

Correctness: Yes

Clarity: Yes, the paper is very clearly written.

Relation to Prior Work: Yes, relation to prior work is nicely discussed and very pedagogic.

Reproducibility: Yes

Additional Feedback: ---EDIT AFTER AUTHOR RESPONSE--- After reading the other reviews and the authors response, my grade remains unchanged, and I think the paper should be accepted.


Review 4

Summary and Contributions: This paper studies bounding causal effects from data collected by randomized experiments contaminated with non-compliance. In particular, the authors assume the instrumental variable (IV) model (Pearl, 2000, Sec 8.2). The authors improve over the existing results by considering a generalized setting where domains of the treatment and outcome are continuous. To address challenges of continuous domains, the authors consider a family of IV models where the functions are parametrized by a linear combination of non-linear basis functions (kernels). The basis functions are presumed to be known while the coefficients are drawn from a multivariate Gaussian distribution. The primary bounding strategy follows the methods of (Balke & Pearl, 1994) (for short, BP94). That is, the authors (1) formulate the causal bounding problem as a series of optimization programs, (2) and obtain the bounds by solving these programs. The authors also discuss some practical considerations for deriving the bounds. For instance, domains are discretized to estimate the observational distribution. A stochastic gradient descent algorithm is employed to solve the formulated computer program.

Strengths: The experiments are really comprehensive. Future work on bounding causal effects in IV models should follow a similar framework. I like the idea of using two coefficients a, b to categorize instances based on the strength of instrument and confounding. The proposed method is verified in each of these categories. I also appreciate the fact that the authors report both positive and negative cases.

Weaknesses: The authors study a critical problem in the causal inference and make some interesting progress. However, I do have some concerns. I am particularly curious about how sensitive the derived bounds with regard to the parametric assumptions of underlying functions. For instance, in the plot of Fig 2, Row 2, Column 1, the actual causal effect seems to lie outside the derived bounds. This suggests that when the instrument is weak and the strength of unobserved confounding is strong, the proposed methods may not lead to valid bounds. Is there any practical method to test the strength of the instrument and confounding from the observational data? Otherwise, this result seems to suggest that the validity of the proposed method relies on untestable parametric assumptions, which makes the significance of this work somewhat limited. On the contrary, the universal partitioning model introduced in (BP94) (i.e., the discretization of the latent space based on the response functions) is robust for any IV models with discrete observed variables. That is, one could always obtain a valid bound in the discrete domains using the linear program formulation of (BP94). This leads to my next question. Is it possible to (1) discrete the observational data into different bins, and (1) obtain a causal bound using (BP94)? It seems that this somewhat naive approach is guaranteed to lead to valid bounds. How does this approach compare to the authors' method? It would be appreciated if the authors could provide some insights.

Correctness: This paper is techinically sound. The empirical methodology is sound and comprehensive.

Clarity: This paper is clearly-written and well-organized.

Relation to Prior Work: The references and discussion of the related work are sufficient.

Reproducibility: Yes

Additional Feedback: -- POST REBUTTAL -- I have read the authors’ responses and other reviewers’ comments. Unfortunately, they did not address my concerns regarding this paper. In particular, the authors claim that the inconsistent bounds in Fig 2 (Row 3, Column 1) is due to issues of finite samples, and “higher sample size will control this error”. This comment is curious since the error in Fig 2(R3, C1) appears to be quite significant. If that is really due to insufficient sample size, the authors may want to develop confidence bounds that control the uncertainties of finite samples. Nevertheless, the insufficient sample size may serve as an alternative explanation. Still, it does not disprove the possibility that the proposed parametric assumptions may be incorrect, which is more likely to be the course of bounding errors in Fig 2. In the end, accepting/rejecting comes down to how general the required parametric assumptions are. While I could see the point of letting practitioners evaluate these assumptions in practice, I am afraid that such a decision may lead to more misuse that it is intended. Due to the nature of causal inference studies, the target causal effect often remains unknown to the investigators. Bounding errors, if they exist, are not likely to be caught during the study. Some investigators may follow up and revisit these assumptions, but there is a chance that their concerns could go unnoticed. Due to these reasons, I would like to maintain my original score. Having said that, I believe this work would be most improved with a discussion on the robustness of the required parametric assumptions. A sensitivity analysis of these assumptions is also encouraged.

[Author Response · NeurIPS 2020]

Thank you reviewers for your time and thoughtful comments! We respond to each reviewer's comments below.

**R1** [Non-convexity issues.] One diagnostic on non-convexity issues is to measure the smoothness of the dose-response

curve. We argue that our experiments demonstrate valid bounds are possible in many cases. We stress that many hard

problems in ML involve non-convex optimization without theoretical guarantees, but these solutions are still useful.

**R1** [Uncertainty intervals for the bounds.] We agree this is important, which is why we mentioned it. There are a lot of

possible ways to do this including links to *(Silva & Evans. 2016)* and *(Fong, Lyddon, & Holmes. 2019)* which is why we

originally thought to leave this for future work. We would be open to including an experiment if the paper is accepted.

**R1** [Choice of functions for $p_\eta$.] Sorry for not being clear. We chose the distribution for $p_\eta$ to be Gaussian purely as a

reasonable example. Ultimately, the choice of $p_\eta$, $p_\eta(\theta|x,z)$, and $f_\theta$ is completely up to the practitioner.

**R1** [Intuition on response.] Our recommendation is that if one does not have any prior knowledge about what the

response function family is, then the most conservative thing is to use a flexible function family such as an MLP basis.

As you point out, this may result in loose bounds as the practitioner provides limited information to constrain the model

space. However, if one does have an idea of the function class of the response (e.g., through expert knowledge, prior

experiments) then one should parameterize $f$ to constrain the considered models and thus obtain tighter bounds.

**R1** [Other non-linear cases.] Thanks, we completely agree with this. We have included an additional non-linear case in

the supplement, but would consider including it in the main text, let us know what you think!

**R1** [Choice of outcome equations.] Our criteria were (a) the simplest choices for both linear (i.e, 2SLS assumptions

satisfied) and non-linear, non-additive cases (i.e., quadratic, interaction term); (b) to keep variance of $Y$ constant for

different settings; (c) obtain "reversal" of the observed vs. true effect as we vary confounding. We have run many more

settings with higher-degree polynomials, trigonometric, and exponential functions, and we plan to add these!

**R2** [Tradeoffs.] Thank you for this question! One way to view the trade-off is via requirements on the bounds: In

some cases, bounding the true effect to be above or below zero could be a valuable insight. In other situations more

details of the causal effect are required to inform decisions, requiring stronger assumptions. So the context can inform

the necessary strength of assumptions. Another view on the trade-off is via domain knowledge: If one has expert

knowledge that an effect is, say, linear, then one can use this to obtain tighter, more informative, bounds. If instead an

expert has information about the smoothness of the causal effect, methods such as regression splines can be controlled

based on this (see Gunsilius, 2019). Our aim is primarily to allow for flexible assumptions. Overall, we argue that a

practitioner is much more likely to have expert knowledge about the form of the response function versus other common

assumptions (e.g., additivity). In these cases they can directly tune the size of the function class to obtain bounds that

are tight enough to suit the particular context at hand. Thank you for asking about this, we will add this to the text.

**R2** [Include Bennet et al., 2019, Lewis & Syrgkanis 2018.] Thanks for these! We will add them to the related work.

**R3** [Not in basis.] Good question. We agree there could be cases where the response is not parameterized by

basis functions. In this case, a practitioner can still use our framework but with a different differentiable response

parameterization. We simply chose basis functions as a flexible example that allows for efficient optimization. For

example, if one has information about the smoothness of the causal effect, methods such as splines/wavelets can be

controlled for this (see Gunsilius, 2019). It's all about allowing for choices as opposed to a one-size-fits-all tool.

**R3** [Lack of prior knowledge.] Agreed, without any prior knowledge, the bounds may not capture the causal effect. This

makes sense as IV cause-effects are unidentifiable from observations without assumptions. We argue that the classic

solution: assuming additivity, is rarely justified by domain knowledge. However, it is not uncommon to have knowledge

about the parametric family or the smoothness of the causal effect, which many causal-effect methods utilize.

**R3** [Grid for Z, relax bound.] This is a deep question. Ultimately, we argue this must be investigated case-by-case. The

more levels of Z the better if we can afford it. In practice, an analyst can choose a few treatment levels and see how the

bounds on the ATEs shrink as more Zs are entered, before setting on a 'satisficing' plateau to use on other ATEs.

**R3** [How constraints baked in.] Thanks. We will add more detail about this in the text.

**R3** [Identifying $p(X|Z)$.] Ah let us clarify this (more in Appendix). We always identify $p(X|Z)$ using grids in "CDF

space": each bin contains the same number of points. This way we don't need to assume any distribution for $p(X|Z)$.

**R5** [Sensitivity w.r.t. response.] We have run many more settings where the guessed response differs from the true

response: high-degree polynomials, trigonometric, and exponential functions, and we'll add these to the Appendix.

**R5** [Causal effect outside bounds.] Curves may miss the bounds as these are estimated bounds, versus population ones.

Higher samples sizes will control this error, but it will be there as in any learning problem.

**R5** [Tests.] There is a degree of testability (e.g., linear models imply falsifiable constraints), but we would rather cover

these issues in a journal version. We want to emphasize that as causal methodologists all we want is to provide tools

that allow for flexible assumption-making, and put the ultimate responsibility where it should be, the domain expert.

**R5** [Discretize, BP94.] Four reasons it won't work: (1) Discretizing $X$ is a non-invertible mapping, so in general

exclusion restriction is destroyed i.e. independence of $Z$ and $Y$ given $\{X, U\}$ stops holding when $X$ is discretized.

(2) If we discretize $X$ and $Y$ each into $k$ categories, the number of response functions and constraints is $O(k^k)$. (3)

"$do(X^* = x^*)$" is ill-defined for a discretization $X^*$ of $X$. (4) Vacuous bounds at $k \to \infty$. Thank you for raising this.

[Meta-Review · NeurIPS 2020]

The work provides a method based on modern machine learning for bounding causal effects under the instrumental variable graph and when both treatment and outcome variables are continuous. Overall, reviewers were positive about the paper, and I share the general assessment, this is a very nice and strong piece of work. Having said that, I will list some serious issues I found when reading the paper (the not so good part), which I expect the authors will take into account and reflect in the camera-ready version of the paper First, the paper’s contribution is overstated, which is not needed due to the high quality of the work (!). For instance, the author says (line 35-36): “In this work, we develop algorithms to compute these bounds on causal effects over all IV models compatible with the data in a general continuous setting. “This is misleading since the work doesn’t consider the most general setting. In particular, assumptions are made about the latent space of the exogenous variables. The assumptions may be reasonable or unreasonable, depending on the context, but they do not solve the most general setting (more below). This issue is exacerbated given the papers says: “One of the major obstacles to trustworthy causal effect estimation with observational data is the reliance on the strong, untestable assumption of no unobserved confounding. “That’s somewhat ironic since the parametrization and corresponding optimization procedure proposed in the paper is valid exactly *because* of such assumptions about the exogenous latent space (!). In other words, the narrative used in the introduction is inaccurate and needs to be improved; the real contribution of the paper needs to be stated more clearly. Furthermore, the comparison with Balke & Pearl, 1994 (henceforth, BP), the canonical result in the field, is misleading since the main strength of BP’s approach avoids imposing *any* parametric constraint over the latent space. In fact, BP is able to do so by constructing a partitioning of the latent space that is universal BUT only works when the endogenous variables are discrete. At first, I thought the paper solved this problem and would offer a counterpart construction for the continuous domain. This was not the case, and the problem was solved by imposing constraints over the latents. There exist a recent attempt to bound continuous effects when Y is continuous, but X is still discrete, which to the best of my knowledge, is also universal as BP’s construction for the discrete case (link: Columbia CausalAI Laboratory, Technical Report (R-61), Zhang and Bareinboim, 2020, https://causalai.net/r61.pdf). I recommend the authors check this result, understand the subtlety involving, and add a short comparison. Again, to the best of my knowledge, it’s not known how to parametrize continuous models in generality, a la BP, when both treatment and outcomes are continuous. It’s okay to add assumptions, but needs to as explicit and transparent as possible about them. Last but not least, the actual causal effect in the simulations (Fig 2, Row 2, Column 1) lie outside the derived bounds, completely off! This issue was explained to Reviewer 5 during the rebuttal stage due to finite samples, but it’s not clear at all in that’s the case. Again, I wouldn’t dismiss the fact that the latent space’s parametrization may be entirely wrong. I tried to avoid using my personal opinion, but I feel it’s extremely dangerous to allow one to impose parametric constraints over the unobservable without any guidance or way of judging its plausibility. Naturally, this wouldn’t happen if the partitioning was universal. Overall, this is a nice piece of work with application in core causal inference and reinforcement learning, therefore, my recommendation is ‘accept.’